# Accounting for forest condition in Europe based on an international statistical standard

Joachim Maes [1,2], Adrián G. Bruzón[3], José I. Barredo [2] ✉, Sara Vallecillo[2], Peter Vogt [2], Inés Marí Rivero[2] & Fernando Santos-Martín[3]

Covering 35% of Europe's land area, forest ecosystems play a crucial role in safeguarding biodiversity and mitigating climate change. Yet, forest degradation continues to undermine key ecosystem services that forests deliver to society. Here we provide a spatially explicit assessment of the condition of forest ecosystems in Europe following a United Nations global statistical standard on ecosystem accounting, adopted in March 2021. We measure forest condition on a scale from 0 to 1, where 0 represents a degraded ecosystem and 1 represents a reference condition based on primary or protected forests. We show that the condition across 44 forest types averaged 0.566 in 2000 and increased to 0.585 in 2018. Forest productivity and connectivity are comparable to levels observed in undisturbed or least disturbed forests. One third of the forest area was subject to declining condition, signalled by a reduction in soil organic carbon, tree cover density and species richness of threatened birds. Our findings suggest that forest ecosystems will need further restoration, improvements in management and an extended period of recovery to approach natural conditions.

Forest ecosystems are a critical component of the world's biodiversity. Yet, deforestation and forest degradation continue to take place in many parts of the world at alarming rates, which contributes significantly to the ongoing loss of biodiversity and increasing effects of climate change[1]. In Europe, forests are expanding[2] and accumulating biomass[3] including deadwood, a proxy for biodiversity. However, several pressures such as eutrophication[4], drought[5], and tree cover loss[1] remain high and continue to undermine the condition of forests. Forest degradation has multiple negative consequences. It results in biodiversity loss, reduces economic output of rural areas and slashes the capacity of forests to deliver ecosystem services such as timber, flood protection, and nature-based recreation[6,7]. Forests ecosystems are Europe's largest terrestrial sink of carbon from the atmosphere[8] contributing significantly to climate change mitigation[9].

Any region's economic competitiveness and security in the long run depends directly on sustainable use of natural resources[10]. Increasing the protection of healthy forests and restoring degraded forests to a favourable condition has thus become an essential objective of the European Green Deal, a policy of the European Union (EU) that couples climate targets to an economic growth strategy. Achieving the double goal of economic growth and sustainability requires going beyond GDP to measure the added value of healthy ecosystems. Ecosystem accounts deliver this necessary statistical framework[11]. In March 2021, the UN Statistical Commission adopted a new global statistical standard, the System of Environmental-Economic Accounting - Ecosystem Accounting (SEEA EA)[12]. SEEA EA is a spatially-based, integrated statistical framework for organizing and tracking biophysical and economic information about ecosystems and it links this information to measures of economic and human activity in a way consistent with the System of National Accounts[11]. Ecosystems are considered as assets and are described through ecosystem extent and condition accounts. Ecosystem assets deliver ecosystem services that are realised within a particular accounting area and supplied to the economy. The SEEA EA provides a unified, international accounting framework for ecosystem condition that is rooted in the concept of ecosystem integrity and practically based on a stepwise approach to

[1]European Commission, Directorate-General for Regional and Urban Policy, Brussels, Belgium. [2]European Commission, Joint Research Centre, Ispra, Italy. [3]Department of Chemical and Environmental Technology, Rey Juan Carlos University, Madrid, Spain. ✉e-mail: Jose.BARREDO@ec.europa.eu

infer the condition of ecosystems[13]. It provides guidance on the selection of variables appropriate to measure ecosystem condition, the choice of a reference condition, and the aggregation of variables into a single condition index.

Here we apply the set of ecosystem accounting rules to map and assess the condition of Europe's forests for two different years, 2000 and 2018. We aggregate seven forest condition variables into a forest condition index to measure the similarity of 44 forest types to a reference condition based on observations in primary and protected forest sites. Our analysis uses regularly updated datasets that describe water availability, soil organic carbon, the number of threatened forest birds, tree cover density, ecosystem productivity, forest connectivity and landscape naturalness. Although there is no shortage of European assessments of forest health[14], condition[15,16] or integrity[17,18], our method is based on an internationally adopted statistical standard. This facilitates acceptance across different social and economic sectors, comparability, and uptake, which are important advantages for decision makers[19]. Our forest condition account provides a consistent framework for the observation, reporting and analysis of past trends and present conditions, can guide investments in the conservation or restoration of degraded ecosystems, and can mainstream the ecological values of forests in policy making and implementation.

This paper serves two purposes. First, it operationalizes a method to account for forest condition based on spatially-explicit, frequently updated datasets of forest ecosystem characteristics. The use of spatial data to assess ecosystem condition enables a large-scale monitoring system with an objective estimation of the area to be considered as in a degraded condition, to establish conservation actions, or to set restoration priorities. Second, this paper provides to our knowledge the first, large scale test of the SEEA EA guidelines on ecosystem condition accounting[12] and presents a forest condition account using most of the European continent as accounting area.

## Results

### Forest condition map

We analysed the condition of 1,964,211 km$^2$ of forest in Europe (Supplementary Table 1). Forest condition had a patchy distribution with high conditions prevailing in the eastern parts of the Alps, the Carpathians, Scandinavia, and along the shores of the Black Sea. The Atlantic plain, the British Isles, and the Iberian Peninsula are characterized by a sparser distribution of forests with a lower condition (Fig. 1a). Across the continent improving forest ecosystem conditions occur locally alongside declining conditions (Fig. 1b). However, most of the forest area in Europe experienced an increasing condition: in 63% of the area the forest condition measured in 2018 was higher than the condition in 2000, although the change is limited to an average of 4.3%. In 37% of the accounting area, the condition in 2018 was lower than the condition in 2000. Forest degradation was more pronounced in north Scandinavia, the Carpathians and the Balkan, the northern Apennines, and in forests throughout the Iberian Peninsula (Fig. 1b). Forest condition declined at a slow rate with an average loss of −3.8% between 2000 and 2018. In 2.8% of the forest area, we observed a loss in condition greater than 10%.

### Condition by forest type

Forest condition varied between 0.31 and 0.78 and mostly increased between 2000 and 2018 (Fig. 2). The growth in forest condition is consistent across most forest types and varied between 0.2 and 14.4%. Using a Mann-Whitney-$U$ test on 1000 randomly sampled grid cells selected to avoid spatial autocorrelation, we show that the average forest condition of 33 forest types was significantly higher in 2018 than in 2000 (Supplementary Table 2). Only the condition of the four Macaronesian forest types occurring on the Azores and Canary islands declined significantly with, on average, 7.5% between 2000 and 2018 (Supplementary Table 2). For seven forest types the changes were not

significant (Supplementary Table 2). Averaging over all the forest types, forest condition was 0.566 in 2000 and increased to 0.585 in 2018, a growth of 1.9%. The upward trend in forest condition is underpinned by increases in each of the seven forest condition variables (Fig. 3, Supplementary Table 3). In Fig. 3, their average values are plotted both against the scale of measurement and the range between the mean lower and mean upper reference level. Figure 3 also provides values that are rescaled between 0 and 1 and which are used to calculate the forest condition index. The average values of vegetation water content, forest productivity, forest connectivity and landscape naturalness are closer to the upper reference level and tend therefore to increase the value of the forest condition index. Species richness of threatened forest birds and tree cover density reached, on average, about 50% of the upper reference level for forest. Soil organic carbon in forests was, on average, only at around one fifth of the upper reference level (Fig. 3) and tends to lower the final value of forest condition index.

### Condition by forest class

We observed lower conditions in transitional woodland and shrub than in broad-leaved, coniferous and mixed forests (Fig. 2). The lower condition in transitional woodland and shrub is mainly caused by a lower tree cover density relative to the reference conditions.

### Forest condition by biogeographical region

The conditions of forests situated in the Black Sea, Alpine, Continental and Boreal scored generally above the European average (Fig. 2). Forests in the Atlantic, Mediterranean, and Macaronesian regions have conditions below the European average. Forests in the Arctic and Steppic regions, naturally characterised by treeless vegetation, recorded the lowest conditions. These regional differences in forest conditions are mainly driven by species richness of threatened forest birds, which weighs more than other indicators in the forest condition index, and which varies more widely (Supplementary Fig. 11). Species richness of threatened forest birds had particularly low scores for the Arctic and the Macaronesian regions.

A SEEA EA compliant forest condition account has been provided in the form of a spreadsheet. The table contains for each forest type the values of the forest condition variables and their corresponding reference levels (see Data availability).

### Sensitivity analysis

The condition index per forest type is accompanied with a parameter sensitivity analysis. The parameters used to calculate the forest condition index are the lower and upper reference levels, which set the scale of each condition variable between a degraded and an undisturbed forest ecosystem, and the weights used to aggregate condition indicators to an index (Table 1). We evaluated the sensitivity of the forest condition index to changes of these parameters by recalculating the index while perturbating the parameters one by one by 10% from their base value. As the sensitivity analysis per forest type produced similar outcomes, we only show the results averaged over all forest types. We found that the forest condition index showed a good stability to the parameters variations as the percentage change remained below 2.5% (Supplementary Fig. 2). In particular, the upper and lower reference levels had a limited influence on the index value. (Supplementary Fig. 2). Changing the weight of soil organic carbon resulted in the highest impact. A 10% increase of the weight lowered the forest condition index with, on average, 8.1%. In contrast, a 10% increase of the weight of landscape naturalness led to a 5.7% increase of index.

### Uncertainty analysis

We evaluated the uncertainty stemming from our choice to use primary and protected forests as the natural reference condition against which the condition of forests has been assessed. Most forests in

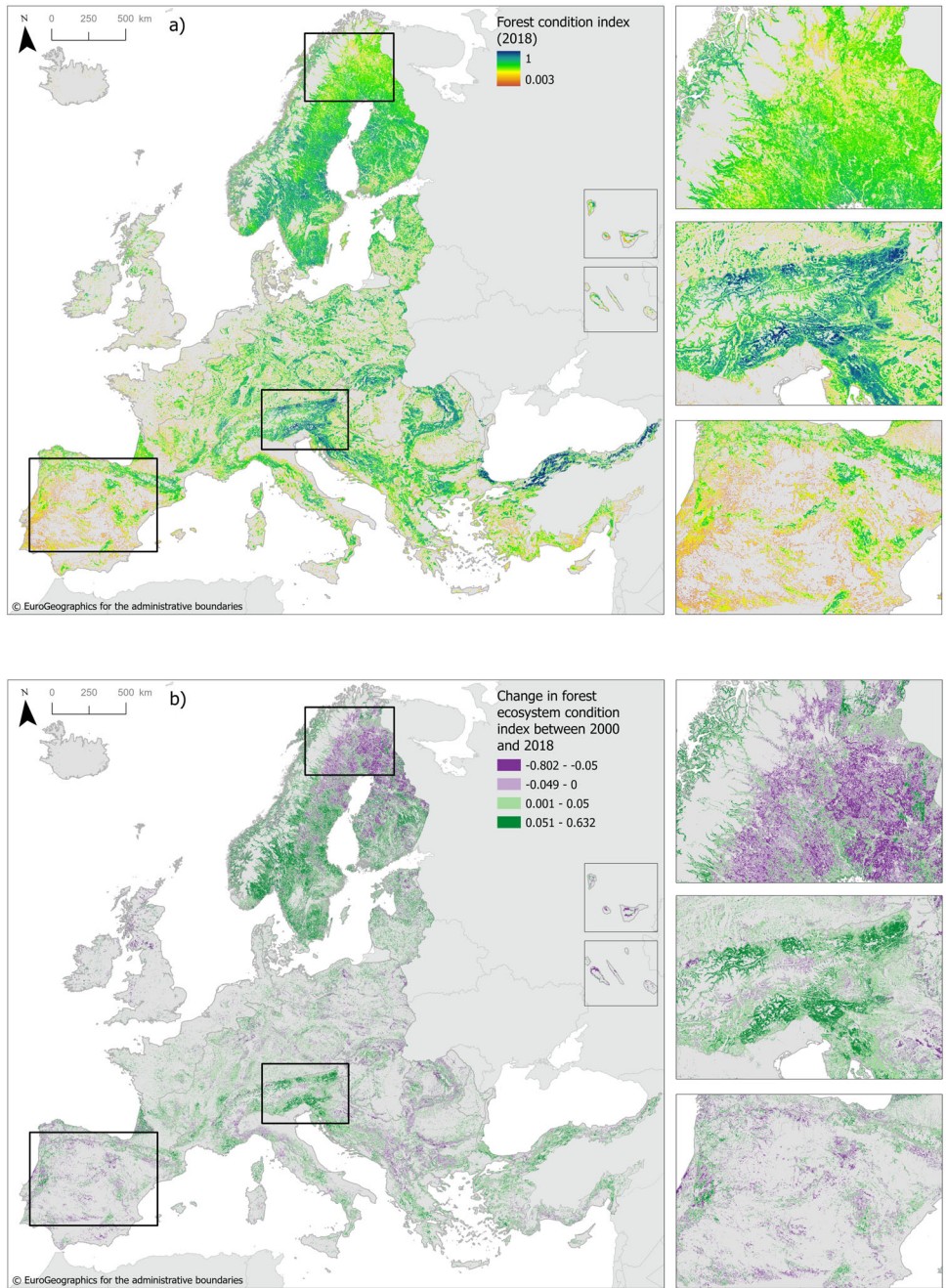

**Fig. 1 | Forest condition in Europe, 2000–2018. (a)** Forest condition map for 2018. **(b)** Change in forest condition between 2000 and 2018. The distribution of the forest condition index by forest type is shown in Supplementary Fig. 1. Insets illustrate changes in Boreal forests, Mediterranean, and Alpine forests. Average forest condition inside the Boreal bounding box changed from 0.625 in 2000 to 0.605 in 2018, a decline of 2%, mainly driven by lower ecosystem productivity (NDVI) and lower soil organic carbon. Average forest condition inside the Alpine bounding box increased from 0.648 in 2000 to 0.682 in 2018, a growth of 3.4% as a result from increases in all condition variables. Average forest condition inside the bounding box covering the Iberian Peninsula decreased slightly from 0.515 in 2000 to 0.512 in 2018 following reductions in tree cover density.

Europe (89%) are considered semi-natural[20]; they consist of secondary vegetation or are plantations while primary forests cover only 3% of the forest area in Europe[21]. To flag this possible bias in our condition assessment, we assigned a semi-quantitative uncertainty level to each forest type using specific criteria (Table 2). We assumed that more uncertainty is introduced if the area of reference sites per forest type is small, if the environmental conditions of the reference site deviate from the average, and if the observed forest class (broad-leaved, coniferous, mixed, transitional woodland and shrub) does not correspond to the expected forest class based on the natural vegetation map of Europe. Forest reference sites exhibit indeed different environmental conditions. The reference sites used in our study are generally situated at higher altitudes and on steeper slopes than non-reference forests (Supplementary Fig. 3). Additionally, reference sites are invariably exposed to a colder average temperature and experience different levels of precipitation (Supplementary Fig. 3). Using forests that are exposed to less favourable climatic conditions to define an upper reference level for condition variables will likely shift the forest condition index of forests under more favourable climate conditions to a higher value. We also found that 72.6% of the forest area in Europe consists of a forest class that corresponds to the natural forest class drawn from the potential natural vegetation map. Inside

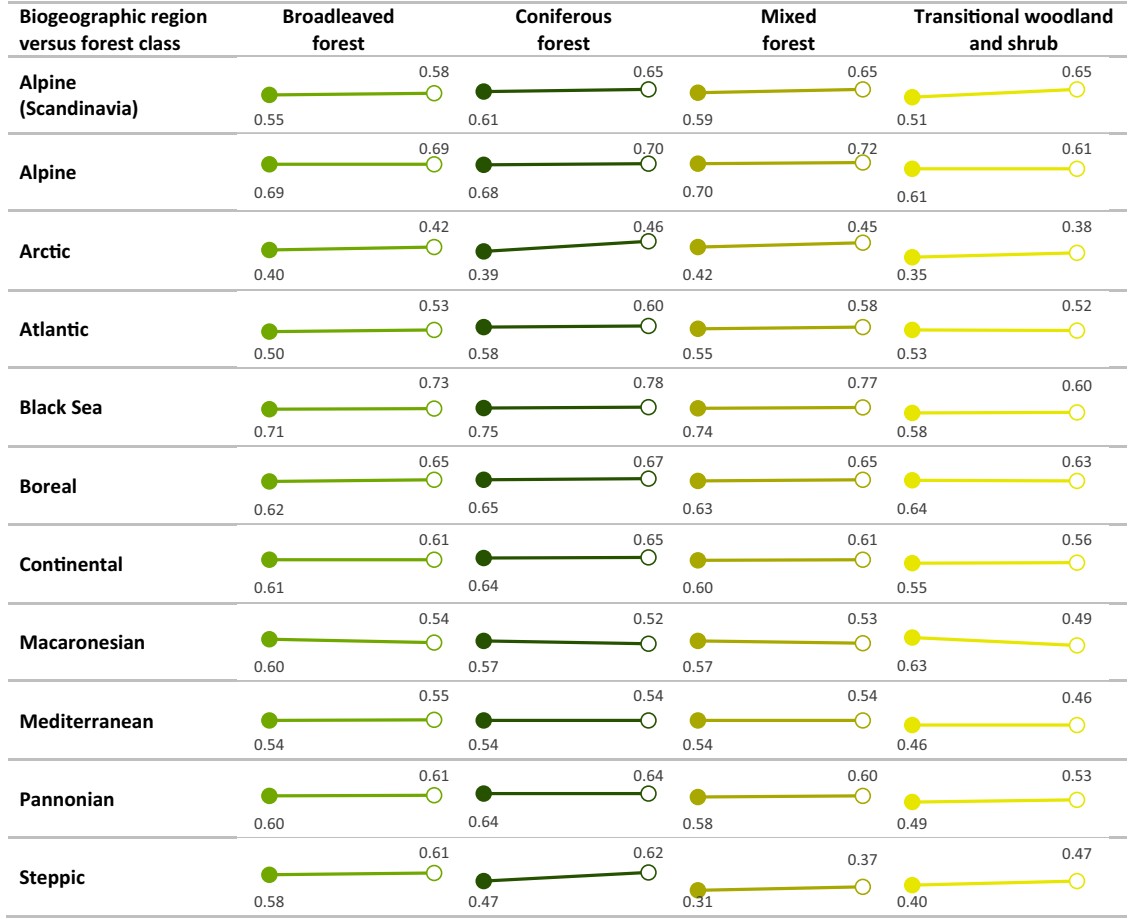

**Fig. 2 | Average condition index per forest ecosystem type on a scale from 0 to 1 for years 2000 (closed circles) and 2018 (open circles).** Forest ecosystem types represent a combination of biogeographic regions and forest classes available in the Corine Land Cover dataset. Results of a Mann-Whitney-*U* test comparing the average condition values between 2000 and 2018 are available in Supplementary Table 2.

reference areas, this share is 78.6%. These data differ widely by forest type (Supplementary Fig. 4). For instance, only 13% of area covered by coniferous forests in the Continental region coincides with a natural forest vegetation type composed of conifer tree species. Arguably, 87% of the observed Continental coniferous forests grow in places where broad-leaved forest is naturally occurring. The result of the uncertainty assessment is a map that assigns four levels of uncertainty to the condition index varying from low to high to the different forest types (Fig. 4). We could assign 25% of the forest area with a low uncertainty level, notably Boreal coniferous and Mediterranean broad-leaved forests. We associated 55% of the forest area with a low to medium uncertainty level, notably Alpine and Atlantic forests, Boreal mixed forests, Continental broad-leaved and mixed forests, and Mediterranean coniferous and transitional forests and shrub. We classified 15% of the forest area under a medium to high uncertainty level, particularly Continental coniferous forests and transitional woodland in shrub in several regions. Our condition estimates for the remaining 5% of forest cover come with a high level of uncertainty including transitional woodland and shrub in the Atlantic and Continental regions and forests in the Black Sea region that are not broad-leaved forests.

## Discussion

Using a UN statistical standard on ecosystem accounting we show that European forests are, on average, in a moderate condition compared to reference conditions found in undisturbed or least-disturbed forest sites. Although the condition of two thirds of the forest area is increasing, change in condition is slow and at many places in Europe offset by declining conditions observed in the

remaining one third of the forest area. Europe's forest ecosystems are productive and relatively well connected to other forests or to the wider, natural landscape. However, the observed distance from the reference state makes clear that forests remain subject to pressures. Enhancing substantially the levels of soil carbon and further conserving and restoring vulnerable species of forest birds will increase the value of the forest ecosystem condition index. Such actions rely on improved forest management and forest restoration. Even then, forests will need an extended period of recovery to approach natural conditions.

European forests are characterised by a high ecosystem productivity, a key ecological variable at the core of numerous ecological processes including decomposition, biomass production, nutrient cycling, and fluxes of nutrients and energy[22]. In turn, these processes determine the capacity of forests to produce ecosystem services such as timber, carbon sequestration and storage, water supply, erosion control, and recreation. Temperature, length of the growing season, water, atmospheric $CO_2$ concentration, and nutrients limit forest productivity[23], explaining differences among Europe's biogeographical regions. Climate change is projected to further impact forest productivity and hence also forest condition[24]. Dryer conditions in the Mediterranean and Circum-Mediterranean regions may compromise productivity and further increase fire hazard. Increasing temperature and growing season are expected to drive forest growth and productivity in the boreal region. From this perspective, the NDVI anomaly or the difference between annual or decadal NDVI and a long-term average, can be considered as condition variable to account for climate change impact on forest ecosystems. Here we opted for a three-year

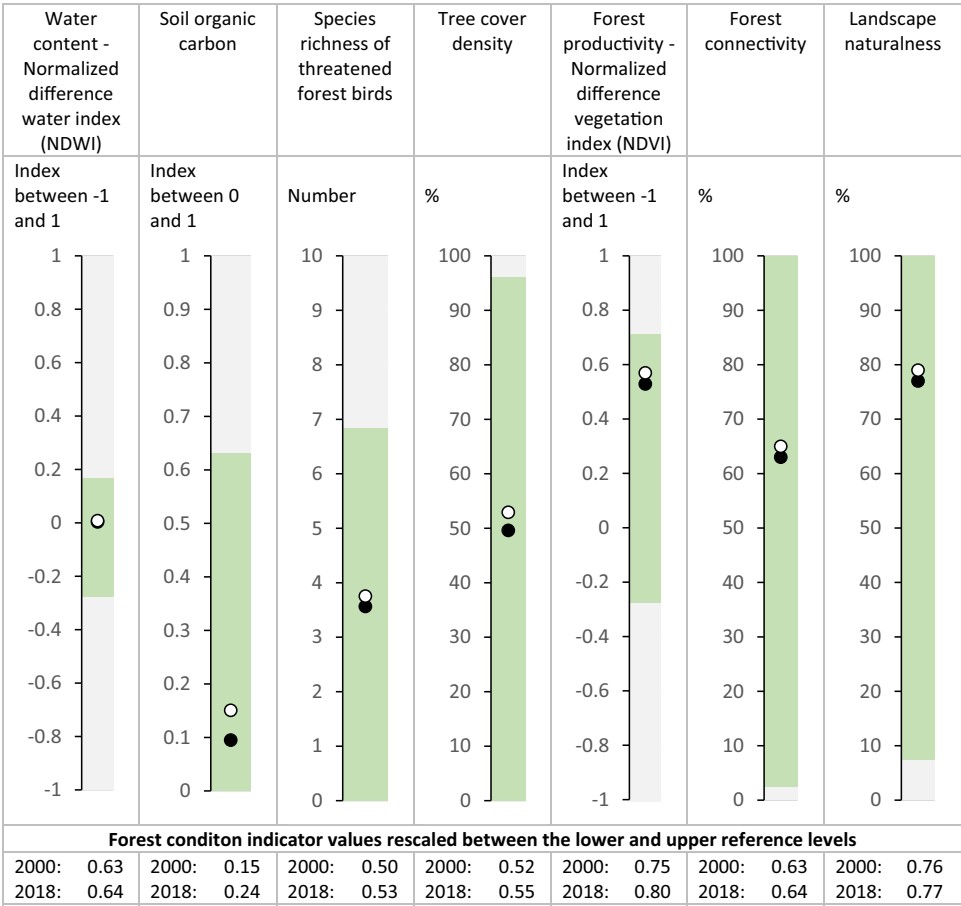

| Forest conditon indicator values rescaled between the lower and upper reference levels | | | | | | |
|---|---|---|---|---|---|---|
| 2000: 0.63 | 2000: 0.15 | 2000: 0.50 | 2000: 0.52 | 2000: 0.75 | 2000: 0.63 | 2000: 0.76 |
| 2018: 0.64 | 2018: 0.24 | 2018: 0.53 | 2018: 0.55 | 2018: 0.80 | 2018: 0.64 | 2018: 0.77 |

**Fig. 3 | European averages for forest condition variables across 44 forest ecosystem types.** Values for 2000 are represented by black dots; values for 2018 by white dots. The green shaded area represents the range between the lower reference levels taken from the ambient distribution of forest condition variables and the upper reference levels observed in reference sites. The grey shaded area represents the measurement scale for each variable. The measurement scale of species richness of threatened forest birds is limited to 10. The data of this figure are available in Supplementary Table 3.

**Table 1 | Forest condition variables assorted per ecosystem condition type (SEEA ECT, System of Environmental Economic Accounting Ecosystem Condition Typology), their possible value range, and their weight in the calculation of the forest condition index**

| SEEA ECT group | SEEA ECT class | Forest condition variable | Abbreviation | Range | Weight in the forest condition index |
|---|---|---|---|---|---|
| Abiotic ecosystem characteristics | Physical state | Vegetation water content - Normalized difference water index (NDWI) | ndwi | −1 to1 | 0.08 |
| | Chemical state | Soil organic carbon | soc | 0 to 1 | 0.12 |
| Biotic ecosystem characteristics | Compositional state | Species richness of threatened forest birds | birds | 0 to 22 | 0.22 |
| | Structural state | Tree cover density | trees | 0 to 100 | 0.21 |
| | Functional state | Forest productivity - Normalized difference vegetation index (NDVI) | ndvi | −1 to 1 | 0.13 |
| Landscape level characteristics | Landscape characteristics | Forest connectivity | fad | 0 to 100 | 0.13 |
| | | Landscape naturalness | lm | 0 to 100 | 0.11 |

average to calculate annual NDVI. Such a window is relatively short to smooth interannual climate variability but it incorporates the impacts of forest fires on ecosystem condition as forests usually need periods longer than three years to recover from wild fires[25,26].

Relative to reference conditions, European forests are underperforming in terms of species richness of threatened forest birds, tree cover density, and soil organic carbon. This provides evidence that European forests are disturbed. Each of these indicators covers a specific aspect of forest condition. Species-based indicators are popular to assess the condition of

ecosystems[27]. We used the species richness of threatened forest birds as an indicator for forest condition where a higher value is associated to a high forest condition. Forests under reference conditions are expected to host twice the number of threatened forest bird species than the number we observed in Europe's forests between 2000 and 2018. Based on local species inventories and modelling, also Newbold et al.[28] found that forest specialist bird species are less likely than non-specialists to occur in secondary forest, forest plantation, cropland and urban habitats, but more likely to occur in primary forest. The bird species

**Table 2 | Criteria used to assess the uncertainty level of forest condition per forest type**

| Criteria | Uncertainty levels (uncertainty scores) | | | |
|---|---|---|---|---|
| | Low uncertainty (1) | Low to medium uncertainty (2) | Medium to high uncertainty (3) | High uncertainty (4) |
| Is the total area of reference sites sufficiently high? | >100 km² and >2% of the total forest area | >100 km² or >2% of the total forest area | <100 km² and <2% of the total forest area | No reference sites |
| Are the reference sites representative with respect to elevation, slope, temperature, and annual rainfall? | z-score reference site <0.3 standard deviations from the mean | z-score reference site between 0.3 and 0.7 standard deviations from the mean | z-score reference site between 0.7 and 1.2 standard deviations from the mean | z-score reference site >1.2 standard deviations from the mean |
| What is the share (%) of the forest area with a naturally occurring forest classᵃ? | >75% | between 50 and 75% | between 25 and 50% | <25% |
| What is the share (%) of the reference area with a naturally occurring forest classᵃ? | >75% | between 50 and 75% | between 25 and 50% | <25% |

ᵃForest classes considered are broad-leaved forest, coniferous forest, mixed forest, and transitional woodland and shrub.

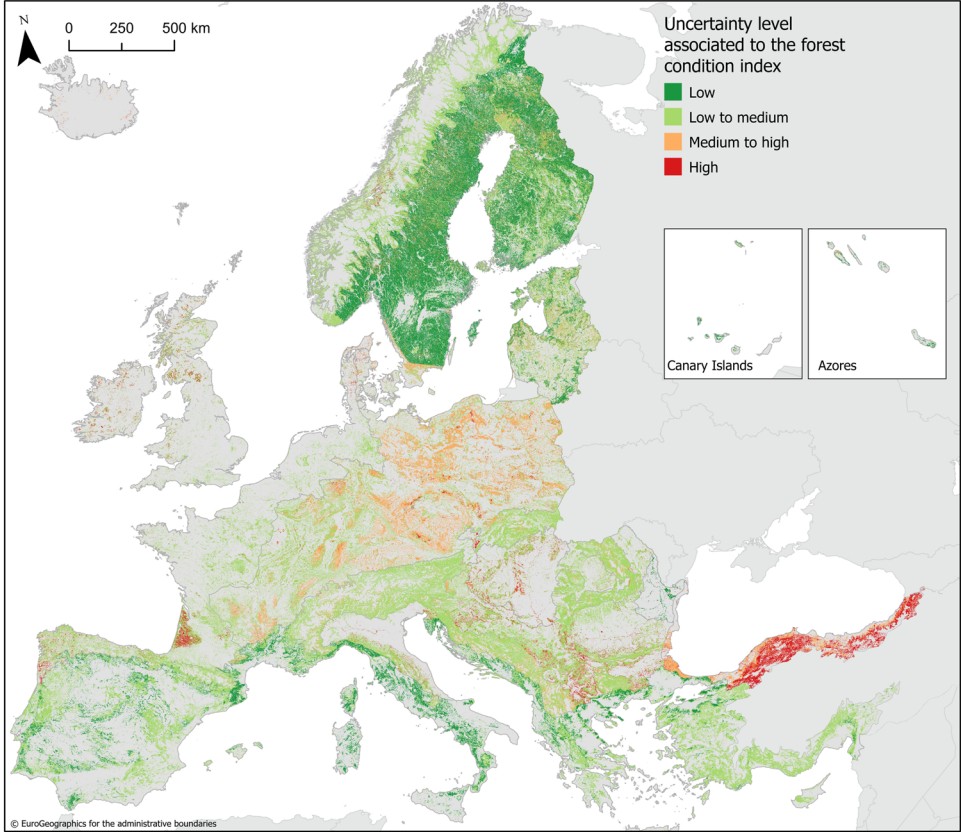

**Fig. 4 | Uncertainty associated to the forest condition index for 2018.** Uncertainty levels are assessed by forest type and mapped using the forest typology map (Supplementary Fig. 7).

retained in our assessment need structurally complex forests with a variety of habitats which are characteristic of forests with old-growth characteristics[29]. For these reasons, this indicator received the highest weight in the design of the forest condition index. The low average values of tree cover density and soil organic carbon suggests that forests are frequently disturbed, now and in the past, corroborating the conclusions of a study mapping forest disturbance regimes[30], those of a study indicating increasing canopy mortality in the last three decades[31], and historical reconstructions indicating major forest changes in Europe. Likewise, the overall use of woody biomass in the EU has increased by 20% between 2000 and 2020[32]. It is probable that variations in forest management in the last decades have contributed to shift disturbance regimes. However, the relative contribution of natural vis-à-vis anthropogenic disturbances in forest condition has yet to be disentangled. Tree cover density can change abruptly due to timber harvesting, forest fires, or tree cover losses from storms or diseases. Based on fine-scale satellite data, tree mortality from both natural and human causes has increased consistently across Europe in the past three decades, with strongest trends in central and eastern Europe and weaker but still positive trends in western and northern Europe[30], areas that coincide with declining forest condition (Fig. 2b). So, too, soil organic carbon can be lost in relatively short periods due to erosion following biomass harvesting, but it takes several decades to replenish the soil[33]. Our assessment also suggests that the role of forests in mitigating climate change can be substantially enhanced.

## Uncertainty and lessons learnt

Data availability limited the inclusion of more relevant variables to assess forest condition than those used in our study. Dead wood[34], forest tree species richness[16], defoliation, tree growth[35], or age structure[36] are frequently cited as indicators for forest health or condition. Yet, they require sampling and modelling to derive spatially explicit variables that cover the entire territory, or they are not collated regularly hindering analysis of trends in forest condition. Where available at national, regional or local level, we recommend using such indicators to monitor forest condition. Through its hierarchical design, the SEEA ecosystem condition typology[37] provides a flexible tool to change, replace or add variables in case better data is available.

We considered exceedance of critical loads for acidification and eutrophication as candidate forest condition variables. In 2016, the critical loads for acidification and eutrophication were exceeded in 30% and 74%, respectively, of forest area in the EU-28[38,39]. However, both indicators are trending negative in Europe. Including these indicators reinforced the upward trend in forest condition between 2000 and 2018. Such a conclusion contrasts with the fact that forest soils accumulated nitrogen and thus still experience the impacts of past and ongoing atmospheric nitrogen deposition[40]. In absence of nitrogen deposition, it still takes decades before forests soils are depleted from excess nitrogen[40] so that eutrophication impacts continue to exert negative impacts on forest condition. From this perspective, the SEEA-EA condition accounts preferably include data on the soil concentration of pollutants, rather than on the pressures that change this concentration.

The choice of indicator weights remains a challenging step in index construction and no weighting system is above critisism[41]. We tested no or equal weights, considered a stakeholder survey to determine weights, but finally opted to use indicator ranking to determine weights. The weights assigned to soil organic carbon and landscape naturalness, and to a lesser extent connectivity and productivity, have a noteworthy impact on the final value of the forest condition index. This should be considered when using the index for making decisions on forest policy, management, or restoration actions.

Changing the reference levels used to scale the forest condition variables between 0 and 1 had a less pronounced influence on the value of the forest condition index than changing the weights. These reference levels were defined per forest type by measuring their maximum value reached in reference sites, which are supposed to be undisturbed primary forests. Not all biogeographical regions have primary forests left and even where they still exist, primary forest may occur in inaccessible places characterised by unfavourable climatic conditions or in small, isolated patches, making them more vulnerable to anthropogenic disturbances and novel natural disturbance regimes such as drought and heat, forest fires, invasive alien species or pest outbreaks. Thus, even by their own standards, forest reference sites can be in low condition. Consequently, we assigned higher uncertainty to the condition estimates for forests covering the Atlantic region, where forest reference sites are scarce, the Black Sea, where data availability limits the identification of reference sites, or the Arctic, where tundra is gradually transformed to shrubland[42] and birch and pine forest[43], also known as borealisation[44]. Between 2000 and 2018 forest area in the Arctic, although still limited in extent, increased by 61%. For the Arctic, we therefore assumed indeed a boreal reference condition which likely explains low forest condition index values.

The Arctic case provides a useful starting point to reflect on the inclusion on the prevalence of naturally occurring forests in the uncertainty assessment. Most forests in Europe consist of secondary vegetation. Our condition accounts demonstrate that these forests are not necessarily of low quality, given our metrics to assess condition. Yet, the ecology of secondary vegetation is different than that of primary vegetation and we acknowledge this by assigning a higher uncertainty to forest types where the forest class deviates from the potential natural vegetation. In almost all biogeographical regions we observe the presence of stands of mainly coniferous forests and transitional woodland and shrub in areas where broad-leaved or mixed forests are expected based on the natural vegetation map of Europe, also in some forest reference sites. For forest types with a high uncertainty level, we recommend using additional data or setting different reference levels. We refrained from including potential natural vegetation as requirement for selecting reference sites. The expansion of forests into the Arctic shows that forests are adapting to changing environmental conditions. So rather than factoring in a fixed map of potential natural vegetation in the selection of reference sites, we recommend carefully observing forest reference conditions, particularly in rapidly transforming biomes. This way, seasonal or annual variability but also long term or irreversible ecosystem changes due to climate change or invasive alien species can be factored in when determining reference levels for ecosystem condition variables. Reference sites can thus be used to determine a dynamic reference condition[45] that can be periodically updated. We call for additional policy initiatives to strictly protect forest reference sites and to restrict management activities to the minimum.

Although we considered the guidance on ecosystem accounting[12] as fairly complete and easy to follow and implement, we have learnt three specific lessons for forest condition accounting. Firstly, the requirement that ecosystem accounts need regularly updated and thus comparable data over time excludes the use of one-off data on forest classification, biodiversity or functional ecosystem characteristics. This omission makes forest condition accounts largely dependent on remote sensing data which can deliver in a biased picture of forest condition. So regular, in situ forest observation is essential to feed forest condition accounts. Secondly, more guidance is needed on how to incorporate forest management in condition accounting instead of treating ecosystems as either natural or modified, for instance by using indicators that can capture forest traits that are sensitive to management, such as tree species composition, age structure or amount of deadwood. Thirdly, primary and old growth forests, irreplaceable as they are, have an extraordinary scientific value for understanding and defining forest reference conditions in the context of global change. This needs dedicated attention in an update of the accounting rules.

## Strengths and applications

Our method can be applied globally to develop forest condition accounts that are compliant with the guidance of the UN's SEEA EA, and hence, with the System of National Accounts, a statistical standard to measure economic activity including gross domestic product. Global land cover data and the International Union for Conservation of Nature (IUCN) global ecosystem typology[46] can be used to assess the extent of forest types. All the forest variables used in this study are available as global datasets or can be globally mapped or modelled. In absence of reference sites, we provide an alternative based on the IUCN's classification of protected areas which is also globally available. Globally available forest extent and condition accounts, in combination with supply use tables quantifying the flows of forest ecosystem services to the economy, would be an important tool to acknowledge the value of forests and to integrate these values in social, economic and environmental policy and decision-making processes.

At European level, the development of the forest condition accounts responds to a clear policy request. The European Green Deal, adopted by the EU in 2019, raised the ambitions for forest protection and restoration, after the European Commission already communicated its ambition to restore forests globally[47]. The European Green Deal is the European Commission's main political guideline that outlines how the EU will transition to a climate-neutral society by 2050. Specific targets for forest ecosystems are included in the EU's biodiversity[48] and forest[49] strategies. Moreover, the EU adopted a

legally binding target on net greenhouse gas removals from land use, land use change and forestry (LULUCF)[50]. Healthy and resilient forests are essential to meet these targets. The most recent policy instrument, adopted by the European Commission on 22 June 2022, is a proposal for a nature restoration law with legally binding targets for the restoration of forest ecosystems[51]. This proposal expects from EU Member States to restore degraded forest ecosystems to a good condition by 2050. An obstacle to the implementation of EU legislation on forest conservation and restoration is the absence of a common method to assess across the board the condition of both protected and unprotected forest ecosystems. Our forest condition accounts provide a first EU-wide baseline that is comparable across countries. They can be used to discriminate between degraded and healthy forests or to define a policy milestone in a forest restoration trajectory towards achieving good forest condition. Finally, the European Commission intends to make the reporting of forest condition mandatory, as part of a larger proposal to amend Regulation (EU) 691/2011 on European environmental economic accounts with ecosystem accounts for several different ecosystem types including forests. Our study provides a first test case at scale which shows that the development and regular reporting of standard forest condition accounts by countries can be achieved using readily available forest datasets.

## Methods

The assessment of forest ecosystem condition followed rigorously the guidelines of the SEEA EA framework[12]. Under this framework, ecosystem condition is defined as the quality of an ecosystem measured in terms of its abiotic and biotic characteristics. Setting up a SEEA EA compliant forest condition requires delineating an accounting area and defining a forest typology, selecting forest condition variables, establishing upper and lower reference levels for these variables, and aggregating these variables to a single value or index. Next, we assessed the uncertainty of the condition index and we ran a parameter sensitivity analysis.

A key criterion for selecting data that underpin accounts is replicability and repeatability. Therefore, the definition of forest ecosystem typology and of different forest condition variables is based on datasets that are regularly updated because of legal requirements or that are part of regular monitoring programs, for instance Copernicus, the EU's initiative on earth observation.

### Definition and delineation of forest ecosystems and description of the accounting area

We used the ecosystem typology developed under Action 5 of the EU's biodiversity strategy to 2020[52] to define forest ecosystems as areas dominated by woody vegetation of various age or they have succession climax vegetation types on most of the area. In practise, this classification is based on the CORINE Land Cover data (CLC), a reference dataset for area-based statistics on land cover and land use in Europe. Our study used the CLC layers 2000 and 2018 and delineated the area of forest ecosystems based on the presence of the following CLC classes: broad-leaved forest, coniferous forest, mixed forest, and transitional woodland and shrub (Supplementary Fig. 5).

We assumed that reference conditions for forest ecosystems in Europe vary depending on biogeographic zones. Therefore, we intersected the four forest classes with 11 biogeographic regions. These include the Alpine, Arctic, Atlantic, Black Sea, Boreal, Continental, Macaronesian, Mediterranean, Pannonian, and Steppic regions. The Alpine region includes all mountainous areas in Europe. Given the strong differences in average temperature and length of the growing season, we decided to analyse forests that occur in the Scandinavian mountains on the border between Norway and Sweden differently than forest in other, more southernly situated mountain chains in Europe (Supplementary Fig. 6). The Anatolian biogeographic region, situated in inner Türkiye, was excluded from this study as it was not

well covered by data and reference conditions. The intersection between four CLC forest classes and 11 regions delivered a forest ecosystem typology consisting of 44 different forest sub-types (Supplementary Fig. 7).

All maps are projected using European Terrestrial Reference System 1989 and Lambert Azimuthal Equal Area projection. This projection takes into consideration the curvature of the earth for representing areas.

### Assessment of forest condition

Ecosystem condition accounts under the SEEA EA record the condition of ecosystems at specific points in time based on a three-step approach. Step 1 defines and selects ecosystem condition variables. Step 2 defines the reference conditions and rescales the variables to ecosystem condition indicators which range between 0 and 1. Step 3 aggregates the indicators into a single ecosystem condition index using indicator-specific weights.

### Selection of forest ecosystem condition variables

We selected the following seven forest condition variables to represent forest condition: vegetation water content, soil organic carbon, species richness of threatened forest birds, tree cover density, forest productivity, forest connectivity, and landscape naturalness (Table 1). This selection stems from a broader list of 59 forest condition indicators that have been agreed with EU member states and experts to map and assess forest condition under Action 5 of the EU Biodiversity Strategy to 2020[53]. The final selection was guided by the SEEA ecosystem condition typology (SEEA ECT) and specific selection criteria[12,37,54]. The SEEA ECT is a hierarchical classification consisting of six classes grouped into three main groups: abiotic, biotic and landscape-level ecosystem characteristics (Table 1). It ensures that the account is built on a parsimonious set of variables that capture the full breath of forest condition. Two other essential selection criteria, besides thematic representativeness, were directionality and data availability. Directionality refers to the requirement that a forest condition variable needs to be related in a straightforward manner to forest condition. It must be sensitive to natural threats or human pressures that decrease condition and to potential restoration measures aimed to increase the condition. Finally, every variable of the condition account needs to be underpinned by spatially explicit and regularly updated data to track changes in the condition that are caused by restoration or human and natural stressors. Next, we provide a description of every forest condition variable including the selection arguments, data sources, the methodology to quantify the variable. The supplementary information contains for every variable a map for the year 2018.

### Vegetation water content - normalized difference water index (NDWI)

Vegetation water content is an important vegetation property which correlates with vegetation health[55]. We used the Normalized Difference Water Index (NDWI) to measure water content and availability in the forest vegetation. Forest ecosystems, and in particular photosynthesis and primary productivity, are limited by water. Low values of NDWI are associated with low content of the forest vegetation canopy and therefore with low forest condition. NDWI ranges in value from −1.0 to 1.0[56] We used three-annual averages to reduce the impact of dry or wet years. NDWI data are available for the period 2001–2019 at a spatial resolution of 30 m. We used the average NDWI from 2001, 2002, and 2003 to approximate the value for 2000. We used the average NDWI from 2017, 2018, and 2019 as value for 2018 (Supplementary Fig. 8).

### Soil organic carbon

Soil organic carbon (SOC) is regularly used as proxy for soil quality[55] or soil health[27,53,57]. SOC concentration, along with its quality and dynamics, is essential to diverse soil functions and ecosystem services.

Soil organic carbon affects the chemical and physical properties of the soil, such as water infiltration ability, moisture holding capacity, nutrient availability, and the biological activity of microorganisms. High values of soil organic carbon (given the soil type) are related with a high forest ecosystem condition. Conversely, a loss of soil organic carbon is considered as environmental degradation. We used the Topsoil Organic Carbon Content for Europe for the year 2003 (OCTOP 2003) at 1 km spatial resolution and soil organic carbon content based on the Land Use and Coverage Area frame Survey (LUCAS) topsoil data 2015. Whereas OCTOP 2003 is a continuous grid map covering most of the forest accounting area used in this study, the LUCAS based data are point samples that required further processing to create a gridded map with a spatial resolution of 1 km. The OCTOP 2003 map does not include data for Iceland and a part of Türkiye. We gapfilled these areas by taking the average SOC per forest type. Data on SOC were not available for the islands that fall inside the Macaronesian region.

We used a Gaussian kriging, available in ESRI ArcGIS to spatially interpolate the LUCAS SOC sampling data. The LUCAS 2015 Soil dataset contains 21,859 unique samples collected in the EU-28, with 1271 samples collected from other countries. The choice for this method was inspired by extensive testing of various kriging methods on an independent dataset[58] to obtain the best possible outcome. The result of this test are included in the Supplementary Table 4. Our results are in line with other interpolations carried out to calculate the value of the SOC in a spatially explicit way[59]. In Yigini and Panagos, P. (2016)[59], three different modelling are compared, obtaining an average correlation of 0.576, while in our case the correlation with an independent data set is 0.56. The OCTOP map for 2003 expresses SOC as a percentage between 0 and 100%. LUCAS soil organic carbon data are expressed in gram per kilogram. To make data comparable, we normalised the values from OCTOP and LUCAS between 0 and 1. We approximated SOC for the year 2000 using the 2003 OCTOP data; for the year 2018, we used the map based on the LUCAS 2015 dataset (Supplementary Fig. 9).

## Species richness of threatened forest birds

Species-based indicators are frequently used to assess the condition of ecosystems[27]. Here we used the species richness of threatened forest birds as a forest condition variable where a higher value is associated to a higher forest condition. The underpinning data are collected by EU countries under Article 12 of the Birds Directive, which protects all wild bird species naturally occurring in the European Union. We made a list of threatened forest birds. Forest species were selected based on the criteria of the European Breeding Bird Atlas (EBBA) that provides a list for boreal and temperate forest birds. To this list, we also added those species associated only to forest ecosystems according to Article 12 data of the Birds Directive. We considered as threatened species those listed in the Annex I of the Birds Directive and species listed under the Red List of Europe. This resulted in a list of 27 species and 9 subspecies. Spatial data on bird occurrence were taken from the first reporting period (2008–2012), available at 10 by 10 km grid cells covering the EU. In this reporting period, three out of the 27 threatened forest species were not observed (Scottish crossbill (*Loxia scotica*), Azores bullfinch (*Pyrrhula murina*), and Krüper's nuthatch (*Sitta krueperi*)). Therefore, species richness is finally based on the occurrence of the following 24 species and nine subspecies which have local distributions: Arctic warbler (*Phylloscopus borealis*), Black woodpecker (*Dryocopus martius*), Brambling (*Fringilla montifringilla*), Collared flycatcher (*Ficedula albicollis*), Corsican nuthatch (*Sitta whiteheadi*), Eurasian eagle-owl (*Bubo bubo*), Eurasian pygmy owl (*Glaucidium passerinum*), Eurasian three-toed woodpecker (*Picoides tridactylus*), European honey buzzard (*Pernis apivorus*), Fieldfare (*Turdus pilaris*), Great grey owl (*Strix nebulosa*), Grey-headed woodpecker (*Picus canus*), Hazel grouse (*Bonasa bonasia*), Middle spotted woodpecker (*Dendrocopos medius*), Northern hawk-owl (*Surnia ulula*), Red kite (*Milvus milvus*), Red-

breasted flycatcher (*Ficedula parva*), Redwing (*Turdus iliacus*), Rustic bunting (*Emberiza rustica*), Semicollared flycatcher (*Ficedula semitorquata*), Tengmalm's owl (*Aegolius funereus*), Ural owl (*Strix uralensis*), Western capercaillie (*Tetrao urogallus*), White-backed woodpecker (*Dendrocopos leucotos*); Subspecies: Coal tit (*Parus ater cypriotes*), Eurasian sparrowhawk (*Accipiter nisus granti*), Fair Isle wren (*Troglodytes fridariensis*), Gran Canaria blue chaffinch (*Fringilla teydea polatzeki*), Great spotted woodpecker (*Dendrocopos major canariensis*), Great spotted woodpecker (*Dendrocopos major thanneri*), Northern goshawk (*Accipiter gentilis arrigonii*), Short-toed tree-creeper (*Certhia brachydactyla dorotheae*), Tenerife blue chaffinch (*Fringilla teydea teydea*).

We chose the first reporting period (2008–2012) of the Birds Directive because it presents a smoothed bias among countries, and it is also more representative of the whole period covered in this study (between 2000 and 2018) for model calibration (Supplementary Fig. 10).

Before spatially modelling the bird species richness, we filtered the raw data to exclude observations that we considered to be of low data quality. As proxy of data quality, we used the total species richness reported including all bird species. Only observations with a total richness at least three species were used in the model. We consider that an observation of three or less bird species in total per grid cell covering an area of 100 km² is not likely nor reliable, suggesting a low sampling effort or poor data quality. Moreover, given the incomplete coverage of reported data for Poland and Romania (Supplementary Fig. 10), and the overall low species richness reported when compared to the neighbouring countries, these two countries were excluded from the model calibration. Data filtering led to the exclusion of 5% of the reported data for the assessment period 2008–2012.

We made a selection of 17 relevant predictor variables[60–62] (Supplementary Table 5) including mean annual temperature, temperature seasonality, annual precipitation, precipitation seasonality, altitude, longitude, latitude, the share of urban land, cropland, forest, shrubland, rivers, sparsely vegetated areas, and grassland, land cover diversity expressed by the Shannon Index for land cover, and the mean and range of the Normalized Difference Vegetation Index (NDVI) in summer[62]. Since the data correspond to a period between 2008 and 2012, as predictors for the model calibration we calculated the mean values between 2006 and 2012 for those variables that were dynamic over time (derived from land cover data or summer NDVI).

Following on a correlation analysis of the 17 predictor variables to assess collinearity, we excluded longitude and latitude from the model due to their high correlation (>0.7) with temperature seasonality and mean annual temperature, respectively.

The dataset used for modelling included a total of 40,575 grid cells with data for threatened forest species richness and the 15 remaining predictor variables. The original dataset was split with 70% of the data used for calibration (28,403 grid cells for the calibration dataset) and 30% used for model validation (12,172 grid cells for the testing dataset). Complementarily, we also performed repeated 10-fold cross-validation using the training dataset, with 100 repetitions.

Based on the calibration dataset, a generalised linear model (GLM) was built using the software R to model bird species richness using a quasi-Poisson distribution (suitable for over-dispersed frequency data). A stepwise forward regression was applied to select variables entering the model. Variables were introduced in the GLM one by one according to the correlation coefficient found between them and bird species richness: from the largest correlation to the lowest. Given that responses to environmental variables are frequently non-linear, the quadratic terms were also included when significant. New variables were gradually added to the model until there was not significant improvement in the proportion of deviance explained (pseudo-$R^2$). Model performance was also assessed using the validation dataset.

The final model to predict species richness for threatened forest birds explains 55% of the deviance (pseudo-$R^2$) and has a root mean standard error (RMSE) of 2.13, showing that model prediction, on average, had a distance of about 2 species from observed richness (Supplementary Table 6). The RMSE of the cross validation was on average 2.13 and with an average standard deviation of 0.03, and average pseudo-$R^2$ of 0.55. These performance parameters indicate that the model used is reasonably robust.

The model validation using the testing dataset produced a similar RMSE of 2.15, and a correlation of 0.74 was found between observed values and predicted values.

We calibrated the model using data for the period 2008-2012. We used the model to map bird species richness for 2000 and 2018 using as predictor variables the share of forest, the share of cropland and the mean summer NDVI for these respective years, while the climatic variables (mean annual temperature and temperature seasonality) were kept constant (Supplementary Fig. 11). Model projections were done at 5 by 5 km, downscaling the original raw data available at 10 by 10 km. The overall pattern of species richness at European level is especially high in North-East Europe, but without reaching the highest latitudes (Supplementary Fig. 11). At a more regional level we found larger values of threatened forest birds especially in mountain areas such as the Carpathians, the Balkan Mountains, the Alps (except at very high altitudes), and the Pyrenees. It is in these areas where forest ecosystems exhibit larger dimensions.

## Tree cover density

Tree cover density (also referred to as tree canopy density) provides information on the proportional canopy coverage per land parcel or grid cell in a range from 0 to 100%. It is used as a condition variable that describes the structure of the forest. Tree cover density data are collected at higher spatial resolution than the Corine Land Cover map used to delineate forests. Hence, tree cover density allows us to assess the structure of the forest inside land grid cells that are delineated as forest in Corine Land Cover. We assume that higher tree cover density relates to a higher forest condition[53]. Loss in tree cover density suggests the effects of tree-removal disturbances and forest degradation. For our analysis, the data at 100 m spatial resolution were used. The data for 2000 are approximated by taking the closest year available (2012).

## Forest productivity - normalized difference vegetation index (NDVI)

Ecosystem productivity is a critical ecological variable considered to be at the core of numerous ecological processes including decomposition, biomass production, nutrient cycling, and fluxes of nutrients and energy[63]. Here we used the normalised difference vegetation index (NDVI) to approximate forest productivity. NDVI is an indicator of photosynthetic capacity of plant canopies. Normalized difference vegetation index (NDVI) is the most frequently used and most well-known vegetation index[64]. Several other vegetation indices are available such as the Vegetation Condition Index (VCI) that compares the current NDVI to the range of values observed in the same period in previous years or the Vegetation Productivity Index (VPI) which assesses the overall vegetation condition by referencing the current value of the NDVI with the long-term statistics for the same period. Clearly, most of these indices use NDVI to assess condition so therefore we opted for using the original NDVI observations to assess forest condition rather than using a derived index. We used NDVI data from the Terra Moderate Resolution Imaging Spectroradiometer (MODIS) Vegetation Indices (MOD13Q1) which have a spatial resolution of 250 m. The data for 2000 are approximated by taking the mean value per grid cell for the NDVI data of the years 2001, 2002, and 2003. The data for 2018 are calculated as the mean of average NDVI in 2017, 2018, and 2019 (Supplementary Fig. 13).

## Forest connectivity

Forest fragmentation is a key aspect in biodiversity, ecosystem services and the ever-increasing pressure from anthropogenic land use. Forest fragmentation may lead to the isolation and loss of species and gene pools, degraded habitat quality, and a reduction in the forest's ability to sustain the natural processes necessary to maintain ecosystem health[65,66]. We measured forest connectivity, the complement to forest fragmentation, of European forests using Forest Area Density (FAD)[67]. FAD measures the spatial integrity of forest land cover and accounts for key fragmentation aspects, such as isolation of small fragments and perforations within compact forest patches. FAD is a landscape variable measured in a local neighbourhood, which is then classified into six degrees of connectivity classes. The result is a map product showing the degree of forest connectivity at 1 ha spatial resolution accompanied by a statistical summary table across the reporting unit (Supplementary Fig. 14). We used the forest typology map (Supplementary Fig. 7) and calculate FAD for each grid cell as the proportion of all forest grid cells within a neighbourhood area with a size of 23 × 23 grid cells or 529 ha, which is centered over the grid cell. This process is repeated for all grid cells resulting in a new map of the same dimensions but showing forest area density values for the analysed neighbourhood of 529 ha over each forest grid cell. This processing scheme (FAD 6-class) is available in the open-source software GuidosToolbox[68].

## Landscape naturalness

The landscape mosaic describes landscape composition or the degree of landscape heterogeneity. The landscape mosaic is derived from the Corine Land Cover map showing 44 land cover classes at a spatial resolution of 1 hectare per grid cell (100 by 100 m) for a series of assessment years over Europe (Supplementary Fig. 15). We aggregated the 38 CLC terrestrial land cover classes into three main land cover types: agriculture, natural and developed. In analogy to forest connectivity, the degree of naturalness is for each grid cell derived by assessing the proportion of natural grid cells within a neighbourhood area of 529 hectares. Finally, the values of naturalness expressed as a proportion between 0 and 100% are grouped into the following twelve categories of naturalness: [100, 95, 85, 75, 65, 55, 45, 35, 25, 15, 5, 0] %. The resulting European map of naturalness in twelve categories is then segmented and analysed per forest type.

Redundancy between the predictor variables was low as all pairwise Pearson's correlations between these variables varied between −0.14 and 0.36 (Supplementary Table 7) except for the two landscape variables, forest connectivity and landscape naturalness (Pearson's correlation coefficient = 0.61, Supplementary Table 7).

## Reference levels

The ecosystem condition variables per forest ecosystem type were rescaled to condition indicators that take values between 0 and 1 to reflect their similarity or distance to a degraded state and a natural reference condition, respectively. We set the minimum value that corresponds to a degraded state of the forest by selecting the minimum variable value based on the ambient distribution within each forest ecosystem type. We used maximum variable values observed in primary forest sites[21] to reflect undisturbed or minimally disturbed conditions. Where primary forests do not longer exist, we identified least-disturbed forest sites if they coincide with a protected area under the most restricted category of IUCN (Ia, Ib and II). In both cases, we retained reference sites only if tree cover loss since 2000 was less than 5%. The year 2000 is used as reference year to measure the reference levels of each variable.

We mapped primary forests using the European Primary Forest Database (EPFD v2.0)[21] and UNESCO's Ancient and Primeval Beech Forests of the Carpathians and Other Regions of Europe (UNEP-WCMC, 2021). The EPFD v2.0 is a GIS database of Europe's known primary forests. The database harmonises data on primary forest from 48 different, mostly field-based, data sets of primary forests, and contains data of 41.1 million ha of primary forest spread across 33 European countries. The UNESCO data set, which is inventoried in the EPFD v2, is disseminated by the custodian due to copyright issues. We merged the polygons of primary forest obtained from the EPFD v2.0 and the UNESCO's data set for creating a unique data set of primary forest.

We used the World Database of Protected Areas (WDPA) (UNEP-WCMC and IUCN, 2020) to map protected forests. The WDPA is a geospatial database of world protected areas grouped in seven categories according to the level of protection and are represented by polygons. For delineating protected forest, we used the categories representing the three highest levels of protection (Ia, Ib and II), corresponding to strict nature reserve, wilderness area, and national park, respectively[69]. These categories represent large natural or near natural areas set aside to protect biodiversity and large-scale ecological processes. Therefore, they represent forest with no active forest management or forest with minimum interventions oriented to conservation objectives.

We calculated forest area and disturbed area in the polygons of primary forest and protected areas using data from Hansen et al.[1] (version 1.8). This data set provides annual maps of tree cover loss (2001-2020) and tree canopy cover (percent of tree canopy cover in 2000) both at 30 m grid cell size. For creating a map of forest and non-forest using tree canopy cover, we used the 20% threshold to classify forest and non-forest[70,71]. The 20% threshold of canopy cover is often considered the limit between open habitats and woodlands[72,73]. Then, we excluded polygons having below 7.5 ha of forest because small forest fragments might provide biased results. We used the 7.5 ha threshold considering that CLC uses a minimum mapping unit of 25 ha and a canopy cover threshold of >30% for delineating forest areas. Therefore, 7.5 ha (i.e. 25 ha × 30%) corresponds to the minimum treed area in a spatial unit of CLC that would be considered forest. In addition, all areas falling outside the forest mask delineated using Corine Land Cover were excluded.

For calculating disturbed area, we created a summary map of tree cover loss in the period 2001–2020. A grid cell exhibiting loss in any year was coded as loss in the summary map. Then, we calculated the total amount of forest, tree cover loss, and proportion of forest loss in the polygons of primary forest and protected areas in the period 2001–2020 in relation to 2000. Polygons exhibiting a proportion of tree cover loss equal or less than 5% of the forest area were considered under low-moderate disturbance activity[74]. Therefore, they were used as reference sites. Polygons exhibiting tree cover loss greater than 5% of forest were excluded. The resulting data set of primary forest contained 2699 polygons, representing two million ha of forest and the data set of protected areas contained 5534 polygons representing 2.5 million ha of forest (Supplementary Table 8, Supplementary Fig. 16).

For the Alpine, Alpine (Scandinavia), Atlantic, Boreal, and Continental biogeographical regions we assumed that the total area of primary forest is sufficient to determine upper reference levels for forest condition variables. For the Macaronesian, Mediterranean, and Pannonian regions we complemented the primary forests with protected areas to enlarge the total area of reference sites for determining upper levels for forest condition variables. Despite providing an alternative for primary forests based on protected areas, not all forest types contain reference sites. For the Arctic, Black Sea and Steppic regions we deemed the total area of primary and protected forest insufficient for determining upper reference levels for forest condition variables. For these three regions we adopted upper levels of forest condition variables from other biogeographic regions, given the forest type. Arctic upper reference levels were taken from the Alpine (Scandinavian) sites; Black Sea and Steppic upper reference levels were taken from the Pannonian reference sites (Supplementary Table 8).

Finally, the forest condition variables were rescaled to forest condition indicators taking values between 0 and 1 using the following formula:

$$I = (V - V_L)/(V_H - V_L) \qquad (1)$$

where $I$ is the value of the indicator, $V$ is the value of the variable, $V_H$ is the upper reference level and $V_L$ is the lower reference level. If the value of the variable is larger than or equal to the high condition value, then the indicator takes value of one. If the values of the variable are smaller than or equal to the low condition value, then the indicator takes value of zero.

## Forest condition index

We aggregated the seven rescaled forest condition indicators to a single forest condition index that takes values between 0 and 1, where 0 stands for a degraded forest ecosystem and 1 stands for a natural, undisturbed forest. We applied weights that increase or reduce the influence of each indicator in the final index. The weights were defined by ranking the seven forest condition variables from 1 (lowest rank) to 7 (highest rank) with respect to five conceptual criteria proposed to select ecosystem condition variables[12,54] (Supplementary Table 9). These criteria are intrinsic relevance or how well a variable reflects ecosystem integrity, instrumental relevance or how well a variable relates to the provision of ecosystem services, directional meaning or the potential for a normative interpretation, sensitivity to human influence, and conformity to the SEEA EA framework. The sum of the ranks was converted to a ratio to define weights for every forest condition indicator so that

$$\sum w_j = 1 \qquad (2)$$

The forest condition index is calculated per forest type at 100 m grid size using a weighed sum of the seven forest condition indicators.

$$\text{Forest condition index} = \frac{\sum_{i=1}^{n}\left(\sum_{j=1}^{7} w_j x_{ij}\right)}{n} \qquad (3)$$

where $w_j$ represents the weight of indicator $j$, $x_{ij}$ is the value of indicator $j$ in grid cell $i$, and $n$ is the total number of grid cells for a given forest type.

The data on soil organic carbon did not cover the entire accounting area. For forest types of the Macaronesian region, soil organic carbon is therefore not used in the forest condition index. For this region, the index is based on the six remaining indicators. The weight of soil organic carbon is for this reason proportionally distributed over the other indicators.

The forest condition indicators have been remapped to an annual composite map at 100 m resolution and were then aggregated into a single map using ESRI ArcGIS Pro version 2.8.

## Uncertainty and parameter sensitivity

The SEEA EA recommends using the natural state as the reference condition[12]. While we adopted this recommendation, it also introduces two elements of uncertainty in the forest account, particularly in Europe where most of the forests consists of secondary vegetation. Firstly, many forest patches in Europe contain a different mix of tree species than can be expected based on the potential natural vegetation. Our accounts evaluate the condition of all forests, including for instance conifer plantations that are grown in areas where broad-leaved forest naturally occur. Comparing the condition of such forest stands to a natural reference condition of coniferous forest given the

biogeographical region introduces bias since the underpinning ecological conditions are different[75]. Secondly, primary or protected forests, here used as reference sites, are often located on less accessible places. Whereas this puts limits to their commercial exploitation for resources including timber, potentially unfavourable environmental conditions may lead to lower reference values for condition variables such as water availability or productivity. In turn, this could systematically bias forest condition to higher values.

We addressed these concerns by performing ex-post a semi-quantitative uncertainty analysis of the forest condition estimates per forest type. This analysis is based on (1) the total area of reference sites relative to the total forest area (2) the representativeness of reference sites based on their average elevation, slope, rainfall and temperature, (3) the share of forest classes that correspond to the potential natural vegetation, and (4) the share of forest classes observed in the reference sites that correspond to the potential natural vegetation. We used Supplementary Table 1 and Supplementary Table 8 to calculate the percentage of reference area relative to the total forest area. We calculated the average elevation and slope of forest types and reference sites using the EU digital elevation model. We calculated mean annual rainfall and mean temperature of forest types and reference sites using the E-OBS gridded dataset version 26e for the period 2000–2020. We used $z$-scores to assess how many standard deviations the average elevation, slope, temperature and annual rainfall observed in reference sites is below or above the overall elevation, slope, temperature and annual rainfall per forest type and calculated a mean $z$-score per forest type for determining the environmental representativeness of reference sites. We used the map of the natural vegetation of Europe[76] to determine the expected forest class (broad-leaved forest, coniferous forest, mixed forest, and transitional woodland shrub). We assigned 0, 1 or more forest classes to each of the 55 level 1 descriptions of potential natural vegetation types that occur on the European continent (Supplementary Table 10). Next, we overlaid our map of forest types with the map of potential natural vegetation and calculated the share of forest area (%) that corresponds to the expected forest class. In a similar way, we assessed the correspondence of forest classes that occur within reference sites with the expected forest classes based on natural vegetation. Using the criteria presented in Table 2, we converted the data per forest type into a semi-quantitative, ordinal scale of uncertainty. We derived a final uncertainty level per forest type by taking an average rounded to the nearest integer assuming an uncertainty score from 1 to 4 (Table 2).

We assessed the sensitivity of the forest condition index to changes in the lower and upper reference levels ($V_L$ and $V_H$ in Eq. (1)) and of the weights ($w_j$ in Eq. (3)). The forest condition index is calculated per forest type and is based on 21 parameters (three parameters, $V_L$, $V_H$, and $w_j$, multiplied by seven indicators). Per forest type, we perturbated one by one the value of each parameter and assessed its deviation from forest condition based on the nominal set of parameters. As most of the lower reference levels are set at their natural low or take zero values, we only increased $V_L$ with 10%. Similarly, we decreased $V_H$ with 10%. We increased the value of the weight with 10% while decreasing the other weights so that the sum of all weights remains equal to 1 (Eq. (2)). Finally, we calculated for every parameter perturbation the forest condition index and calculated its percentage deviation of the nominal index value.

### Reporting summary
Further information on research design is available in the Nature Portfolio Reporting Summary linked to this article.

## Data availability
We used the following data in this study. The forest ecosystem typology is based on Corine Land Cover https://land.copernicus.eu/pan-european/corine-land-cover and the distribution of biogeographical regions of Europe https://www.eea.europa.eu/data-and-maps/data/biogeographical-regions-europe-3. Vegetation water content - Normalized difference water index (NDWI) is available at https://developers.google.com/earth-engine/datasets/catalog/LANDSAT_LC08_C01_T1_8DAY_NDWI. We used the Topsoil Organic Carbon Content for Europe for the year 2003 (OCTOP 2003) resolution by the European Soil Data Centre (ESDAC) of the Joint Research Centre https://esdac.jrc.ec.europa.eu/content/octop-topsoil-organic-carbon-content-europe and the soil organic carbon content based on the Land Use and Coverage Area frame Survey (LUCAS) topsoil data 2015; https://esdac.jrc.ec.europa.eu/content/lucas2015-topsoil-data). The species richness of threatened forest birds was modelled based on the following datasets: bird species distribution maps collected under Article 12 of the EU Birds directive as dependent variable https://sdi.eea.europa.eu/catalogue/srv/eng/catalog.search#/metadata/7c2dd14f-60b6-4009-aca8-5d20300479a9; climate data (annual mean temperature, temperature seasonality, annual precipitation, precipitation seasonality) https://www.worldclim.org/data/worldclim21.html; Corine land cover data, and NDVI for summer: https://developers.google.com/earth-engine/datasets/catalog/MODIS_MCD43A4_006_NDVI. Data on tree cover density are available from the Copernicus Land Monitoring Service for the years 2012, 2015 and 2018 (Supplementary Fig. 12). https://land.copernicus.eu/pan-european/high-resolution-layers/forests/tree-cover-density. The data for forest productivity - Normalized difference vegetation index (NDVI) are available at https://lpdaac.usgs.gov/products/mod13q1v006/. Data on primary forests were sourced from the European Primary Forest Database (EPFD v2.0)[21] and UNESCO's Ancient and Primeval Beech Forests of the Carpathians and Other Regions of Europe available at https://www.protectedplanet.net/903141). We used the World Database of Protected Areas (WDPA) (UNEP-WCMC and IUCN, 2020, www.protectedplanet.net) to map protected forests. The map of tree cover loss is available https://earthenginepartners.appspot.com/science-2013-global-forest. The EU digital elevation model is provided by Eurostat: https://ec.europa.eu/eurostat/web/gisco/geodata/reference-data/elevation/eu-dem. The E-OBS gridded data are available at: https://surfobs.climate.copernicus.eu/dataaccess/access_eobs.php#datafiles. The map of the potential natural vegetation of the European continent can be accessed here https://www.synbiosys.alterra.nl/eurovegmap/. The administrative boundaries used for the maps can be downloaded at https://ec.europa.eu/eurostat/web/gisco/geodata/reference-data/administrative-units-statistical-units. We generated the following datasets: the forest typology map, the maps of the forest condition 2000 and 2018, the uncertainty map, and the forest extent and condition accounting tables. These datasets have been deposited in Zenodo: https://doi.org/10.5281/zenodo.7741636.

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

## Acknowledgements

The views and opinions expressed in this article are those of the authors and do not necessarily reflect the official position of the European Commission. We thank Giovanni Caudullo for technical support. This research was partially supported by EU Horizon 2021 grant project 101060415 — SELINA (Science for Evidence-based and sustainabLe decIsions about NAtural capital) (A.G.B. and F.S.-M.).

## Author contributions

F.S.-M., J.M., J.I.B., S.V., and P.V. designed the study. A.G.B. and F.S.-M. analysed the data. A.G.B. mapped the forest condition index. J.I.B. and I.M.R. analysed the forest reference sites. S.V. analysed the distribution of forest birds. P.V. analysed forest connectivity and landscape naturalness. J.M. performed the sensitivity and uncertainty analysis and led the writing of the manuscript. All authors wrote and commented on the manuscript.

## Competing interests

The authors declare no competing interests.
