## [Peer Review File · Nature Communications]

REVIEWER COMMENTS

Reviewer #1 (Remarks to the Author):

The paper applies a global statistical standard on ecosystem accounting to assess the condition of European forests in a spatially-explicit fashion. The authors combine seven metrics (water availability, soil organic carbon, tree cover, productivity, connectivity, naturalness, threatened species biodiversity) into a weighted index 0-1, which they calibrate using primary\least disturbed forests as a benchmark. Calibration is done separately for each forest type (broadleaf, conifer, mixed, transitional forest), and biogeographic region combination (n=44). The spatial results are then compared to country-level reports on the conservation status of European forests, in order to define an index threshold discriminating between habitat in good vs bad conservation status.

Overall, this is a well-written paper and a noteworthy piece of spatial modelling. The reference to the system of Environmental-Economic Accounting – Ecosystem accounting (SEEA-EA) is particularly relevant. As a reader not familiar with this system, however, I would have appreciated additional details on how this approach expands and improves ‘traditional research’ on habitat integrity\naturalness in practice. Multi-parametric indices have often been used on these regards, and there is abundant literature on the issue (see for instance the seminal paper by McElhinny et al. 2006 - <https://www.sciencedirect.com/science/article/pii/S0378112706005123> , not cited in this paper). This aside, the topic tackled remains a particularly relevant and timely one.

I have two main criticisms about this paper.

First, while I find reasonable the idea of using primary forests, or at least strictly protected forests, as a reference, I am afraid the manuscript does not account properly for the biased distribution of these forest patches. In many parts of Europe, primary forests (but a similar reasoning holds for strictly protected areas) occur in remote areas with low productivity and low suitability for agriculture, either for being in rugged mountain areas, or for being in waterlogged swampy areas. The authors do disentangle their data among biogeographic regions\forest types combination, but I am afraid this is not enough, since reference forests are probably not representative of the overall forest area WITHIN each region\type combination. For instance, if most primary forests in a region\type combination occur on rugged, steep terrains, the reference values for Soil Organic Carbon or productivity might be well below the average for that region. This would overestimate the forest condition of impacted forests being located in more favourable environmental conditions. A possible solution could be using a finer distinction of forest types. The EPFD itself uses the 13(+1)-classes classification developed by the EEA, which are spatialized based on the map of Potential Natural Vegetation by Bonn et al. 2002. I provided a simplified spatial version of the PNV map in my 2020 paper in DDI (Sabatini et al. 2020 – Diversity and Distribution). Even if we cannot be sure a given pixel of deciduous forest actually belongs to the respective PNV class, limiting the comparison to forests within the same PNV class within each

region\type combination ensures that the forests being compared have similar environmental conditions (if not composition), therefore strongly reducing the bias highlighted above. While I see the practical advantage of using Corine Land Cover classes, a simple broadleaf vs conifer distinction is way too coarse to account for the ecological differences between (extremely) different growing conditions.

My second point is to a certain extent related to the first. It relates to the way plantations and forests composed by non-native species are considered. At the time being, the ms simply distinguishes between conifer and deciduous forests, not questioning whether the composition of the forest is coherent with the biogeographical context and PNV. For instance, conifer plantations of non-native species are widespread in central Europe. By simply comparing one of these stands to a reference conifer forest in the same biogeographical region there is the risk of returning an overoptimistic picture of the state of these plantations. They might be in 'good' conditions when it comes to soil organic carbon, tree cover and so on. But the appropriate reference for that stand is not a native conifer forest (which is likely located in very different ecological conditions), but rather the broadleaf forest that should be there, if we did not replace it with conifers. This makes a big difference in terms of productivity, biodiversity, SOC and so on. Whether it is relatively easy to double check whether a coniferous forest occurs where the PNV is actually broadleaf forest, plantations of broadleaf forests (e.g., Eucalyptus forests) growing where the PNV is, for instance, an oak forest (e.g., Northern Spain) might be harder to discriminate. Yet, I have the impression missing this crucial point might give the false impression that forests in certain regions (e.g., central Europe) are in a much better condition than they actually are. The super positive results obtained for Austria (one of the countries with the highest Fellings as percent of net annual increment – 87.1% based on Indicator 3.1 of Forest Europe 2020).

Some additional points. As a reader, I found the methods difficult to follow, due to the fragmentation between the main text and SI. Honestly, I would give a more thorough description of the data and methods in the main text, and would only refer the reader to the SI for those predictors for which substantial modelling\pre-processing was necessary (e.g., the SOC or biodiversity modelling). The methods part would results substantially longer, but this would also reduce redundancy (some paragraphs are repeated both in the main text and SI).

I don't quite understand the need of section 1 in the SI. Why making the philosophical distinction between forests and forest ecosystems if, due to data constraints (CORINE), forest was treated simply as a land cover class? Is this section really needed?

Soil Organic Matter – Please provide more details on how the kriging technique was performed, and the goodness-of-fit evaluated. You state: 'The choice for this method was inspired by extensive testing of various kriging methods on an independent dataset¹² to obtain the best possible outcome. Our results are in line with other interpolations carried out to calculate the value of the SOC in a spatially explicit way¹³.' I find this statement unconvincing and not transparent. Make sure you give info on the total number of LUCAS soil plots for 2015. Why mentioning the 2009 LUCAS data if not used?

Threatened birds – I’m puzzled by the lack of a spatial component in the model. Ploton et al. 2020 (<https://www.nature.com/articles/s41467-020-18321-y>) explains very well the risk of neglecting the spatial component in these kind of models. Including a spatial component to the GLM would probably change the relative importance of different predictors. I also recommend using spatial cross-validation when checking for model performance.

Weights for the metrics – Attributing weights is a crucial step of any multi-parametric index. McElhinny et al. 2006 provides some guidance, but there is no real way of ensuring the weights aren’t, to a certain extent, arbitrary. I feel the manuscript should be much more transparent on this, and explain how crucial this step is in the main text, together with a discussion of the strengths and limits of the approach chosen.

Line-to-line comments:

TITLE – From the title alone it’s unclear what ‘standard’ the authors are referring to. What about replacing ‘global statistical standard’ with ‘global statistical accounting standard’

L42 – “Above 25%” change to “more than 25%”

L42-44 – I’d include other examples of degradation beside defoliation (e.g., soil erosion, biodiversity loss, decreased resilience to disturbance).

L63 – “The conditions to develop such a method are ideal”. Unclear. Please rephrase.

L95 – It should be ‘Increase by 95%’, I think (I’m not a native speaker)

L100 – I think a simple t-test is inadequate given the intrinsically spatial (=autocorrelated) nature of the data. A GLMM with a spatial component is probably more adequate here.

L100 “than in other the forest classes” . Delete ‘the’

L117 – “A SEEA EA compliant forest condition account...”. Convolved sentence. Please revise.

L148-149 – “(average forest condition 2018 = 0.53)”. Unclear whether this refers to ‘Mediterranean forests’. If yes, why not reporting also the average for the other forest types listed? How do these values relate to those reported at L154-155.

L204 – ‘well connected’ – Honestly, I don’t find a FAD metric a very convincing indicator of connectivity. In the same 23x23 moving window, the same amount of forest pixels might be interspersed and disconnected, or all clustered in a connected, large patch, without the FAD distinguishing the two situations. I would avoid overselling these results, and maybe delete the whole paragraph L204-208.

L219 – “...productivity in the boreal.” Add ‘forest’ after ‘boreal’.

L253 – The MAIN limitation, i.e., the low representativity of the reference dataset for the forests in each region\forest type combination is not even mentioned here.

I hope this helps

Francesco Maria Sabatini

Reviewer #2 (Remarks to the Author):

The study by Maes et al. examines the condition of forests in Europe and investigates the changes from 2000 to 2018. For this purpose, an index was determined, which is based on seven ecosystem variables like tree cover density. The authors have prepared these variables for the whole of Europe, mainly from established satellite products, with great effort.

However, it is not entirely clear to me what is really novel and innovative about this study. It seems that the basic method has already been published. So, this study is a pure application of a known method to all forests of Europe. It must be mentioned that similar products have already been published by other scientists (which, to be fair, the authors themselves also state L77-78, L80-82). I would recommend to highlight more what is special about your study, besides mentioning 'operationalizes a method' (L83) and 'first, large scale test' (L89). The methods are clearly described and the figures are mostly comprehensible. The statistics for the calculation are rather simple, which I very much welcome, because it makes the calculation easy to understand.

Detailed comments:

- The title and text often refer to "global statistical standard". What is that supposed to be? This is not a common term in statistics. For the title it is maybe not needed to mention.

- The introduction is well written (perhaps a bit too long), however the references are mainly reports or book chapters (ref 1-3,6,8,10-18). These are sometimes not clear to find (no links were given) and it is not clear to me if these are peer-reviewed. I find this unusual. I would like to see more state-of-the-art literature, which is definitely listed later in the Discussion.

- It is not clear in the main text what the Europe-wide forest condition index is based on. It is described in detail in the appendix. However, it would be helpful to have a sense of what data goes into the index before reading the results. This should be mentioned briefly (L84 'spatial data').

- In addition, the selection of the reference levels (min and max) has a strong influence on the final index. It would be advisable to see some sort of sensitivity analysis on the assumptions made. However, it should not affect the analysis of the changes.

- L97-L101: I find the temporal changes in the index (from 0.57 in 2000 to 0.58 in 2018) to be very small. I was surprised that this change should be significant, considering the high standard deviations of the indicator values given in Table 1 (last two columns – SD between 0.12 and 0.19).

- L103-104: 'indicators ... between 0.6 and 0.8'. Where do I see that? Table 1 shows indicator values between 0.2 and 0.8. Right?

- L105: species richness reached 0.48 in 2018 (not 0.52), right?

- L108-113: Where do I see the results to these statements, in Figure 1?

- Table 1: How have you calculated the indicator values? For example, SOC: the reference levels are 0 – 0.63. If transforming the 2000-value 0.09 it should be 14.3% and not 20%. Or do I have a mistake in thinking?

- Table 1: I would recommend to add the final forest condition index in this table including also the sd.

- Figure 3: I'm really sorry, but I don't fully understand this figure. What do the dots stand for? Are they individual countries or regions?

- L209: highest indicator value for 2018 in NDVI. I am surprised, because 2018 was particularly dry and therefore the forests probably less productive. What has this high indicator value then for an interpretation?

Reviewer #3 (Remarks to the Author):

General comments

The article presents a comprehensive assessment of ecosystem condition of European forests based on SEEA EA methodology. Without any doubt, the study provides valuable contribution to the application and development of SEEA EA. European states will be increasingly searching inspiration and methodological guidance for condition assessment in ecosystem accounting.

As authors admit, there is available relatively extensive scholarship on the assessment of forest condition in Europe. The major innovation is computation of forest condition index based on seven condition variables. However, the procedure of selection and calculation of the index would require further attention.

Especially regarding the description of assigning the weights and uncertainty analysis of the index. While authors declare that the change of the condition from 0.57 to 0.58 in the period 2000 – 2018 is statistically significant, the uncertainty range associated with this change is not reported.

The general comment related to the System of Environmental-Economic Accounting – Ecosystem Accounting (SEEA-EA). Authors are focusing primarily on ecosystem condition but present also data on ecosystem extent. As the SEEA EA system consists of several mutually related types of accounts, including extent, condition, ecosystem services supply (biophysical and monetary) and monetary asset account, authors should better position their work in this framework. Rather than technical discussion of particular parameters, it would be beneficial to discuss more context-relevant possibilities for application of condition assessments at the European level and national states.

Authors selected a top-down indicators approach based on European-wide datasets. Therefore, the study builds on available data to support SEEA Ecosystem Accounting. Moreover, condition assessment is based on the distance to the reference condition, i.e. primary intact or protected forest. What is the theoretical framework for the assumption of the good condition only in a natural state? What is the role of the forest management for the condition status when delivering various ecosystem services? These important questions has not been addressed by authors.

Related is the logic behind the indicators selection and weighting. The selection of indicators and assigning of the weights is not sufficiently substantiated by authors. For the analysis to be sufficiently transparent, this should be explicitly addressed and described.

Finally, authors apply SEEA EA approach. What are the lessons learnt and recommendation for future applications of ecosystem condition accounting? Authors list strengths connected to nature restoration targets and global analysis, but it could be interesting to see recommendations or evaluation for the SEEA EA accounting itself. Are there some obstacles or what the future directions are?

Specific comments for the article (line no. indicated):

Title: Authors are analysing a condition and its change in European forests. It would be more accurate to incorporate change into the title. Maybe a focus just on recent period (2018) would lead to more straightforward analysis than comparing only 2 terms.

91-92: Authors are focusing on ecosystem condition but here they report changes in ecosystem extent. Ecosystem extent changes are presented in Supplement. It would be good to align extent results with the whole article, either incorporate in main text or move completely to Supplement. (Also in the context of extensive discussion of extent change in lines 192-203).

97-100: Average ecosystem condition is presented without uncertainty range, so the reader is not guided what is the possible range of the index.

150: Spatial mean condition is 0.622. While average condition for forest types is 0.58. It is clear that this is the outcome of computation approach, but authors could maybe report this in main results and/or align these two numbers.

158: Comparing condition with Habitats condition: this part looks disconnected to the overall assessment. It could be one of the main outputs, but this would require restructuring the article and elaborating more on the prerequisites of the assessment.

161-164: What was the aim of determining a threshold value? It is not fully understandable, how the threshold values was derived.

204 – 205: Repetition of statement from lines 185-187.

254: Selection of forest condition indicators. Authors give a rationale for the overall selection of indicators measuring ecosystem condition. They selected 7 indicators. However, some of the commonly used indicators are intentionally excluded. For example, defoliation or common forest bird species. Selection of indicators influences final index values and the sensitivity to indicator selection is not known. That's one of the problematic points of the study. What is the effect of other indicators? If other metrics were included, would be overall condition index different? Authors illustrate this by critical loads exceedance but this discussion is not generally applicable (of course, it would be preferable to use state rather than pressure indicators).

300: Authors state that presented method can help member states report on Habitats Directive. However, as this method was developed to support SEEA EA, I would appreciate elaboration of this proposal with regard to current reporting obligations. Would really member states report ecosystem condition in the same way within SEEA EA and Habitats Directive? See also the point above.

333: Authors use intersection of Corine land cover and biogeographic regions for the classification of forest types. This approach provides a meaningful overview at the European level. However, it would be beneficial to discuss implications of this approach for the assessment of forest condition with regard to other classifications.

395: Partial repetition of text from lines 160 – 170.

Supplementary information

58 – 66: Forest typology was derived from the intersection of Corine Land Cover and biogeographic regions. The resulting classification reflects broader regions. However, it is not able to discern specific habitat types. For example, continental broad-leaved forests still incorporate large diversity of forest stands, regarding species composition, forest types and management. How this relatively coarse approach could influence resulting condition values?

94: The methodology behind the selection of indicators is not explicitly described. What was the exact selection procedure? Literature search, expert opinion, etc.? This should be transparently described.

429- Table S7: The same remark can apply to weight selection. How were weights determined, respectively how was the importance of the indicators assessed?

451: When comparing condition index to EU Habitats status, authors should discuss the data entering this assessment. To which extent it's harmonized among member states and what types of measurements are entering this assessment?

Reviewer comments

responses in blue font color

Reviewer #1 (Remarks to the Author):

The paper applies a global statistical standard on ecosystem accounting to assess the condition of European forests in a spatially-explicit fashion. The authors combine seven metrics (water availability, soil organic carbon, tree cover, productivity, connectivity, naturalness, threatened species biodiversity) into a weighted index 0-1, which they calibrate using primary\least disturbed forests as a benchmark. Calibration is done separately for each forest type (broadleaf, conifer, mixed, transitional forest), and biogeographic region combination (n=44). The spatial results are then compared to country-level reports on the conservation status of European forests, in order to define an index threshold discriminating between habitat in good vs bad conservation status.

Overall, this is a well-written paper and a noteworthy piece of spatial modelling. The reference to the system of Environmental-Economic Accounting – Ecosystem accounting (SEEA-EA) is particularly relevant. As a reader not familiar with this system, however, I would have appreciated additional details on how this approach expands and improves ‘traditional research’ on habitat integrity\naturalness in practice. Multi-parametric indices have often been used on these regards, and there is abundant literature on the issue (see for instance the seminal paper by McElhinny et al. 2006 - <https://www.sciencedirect.com/science/article/pii/S0378112706005123> , not cited in this paper). This aside, the topic tackled remains a particularly relevant and timely one.

The System of Environmental-Economic Accounting – Ecosystem accounting is indeed a fairly new development. It was adopted as an international statistical standard by UNSDC in March 2021 and it aims to mainstream biodiversity and ecosystems in national statistical data collections which are used for official reporting. So, whereas our work on forest condition is indeed founded on earlier research on ecosystem integrity and naturalness, we argue that the added value of our contribution mainly comes from adopting the accounting framework that in the context of the EU ecosystem condition accounts will be mandatory on 2024. It is a standard that convinces statistical offices and governments about data quality while integration of ecosystem data in national accounting systems greatly enhances uptake and awareness across different economic and societal sectors. We highlight the value of accounting in the introduction L34-L50, L59-L62 and in the discussion under the section Strengths and applications (L311 and further).

I have two main criticisms about this paper.

First, while I find reasonable the idea of using primary forests, or at least strictly protected forests, as a reference, I am afraid the manuscript does not account properly for the biased distribution of these forest patches. In many parts of Europe, primary forests (but a similar reasoning holds for strictly protected areas) occur in remote areas with low productivity and low suitability for agriculture, either for being in rugged mountain areas, or for being in waterlogged swampy areas. The authors do disentangle their data among biogeographic regions\forest types combination, but I am afraid this is not enough, since reference forests are probably not representative of the overall forest area WITHIN each region\type combination. For instance, if most primary forests in a region\type combination occur on rugged, steep terrains, the reference values for Soil Organic Carbon or productivity might be well below

the average for that region. This would overestimate the forest condition of impacted forests being located in more favourable environmental conditions. A possible solution could be using a finer distinction of forest types. The EPFD itself uses the 13(+1)-classes classification developed by the EEA, which are spatialized based on the map of Potential Natural Vegetation by Bonn et al. 2002. I provided a simplified spatial version of the PNV map in my 2020 paper in DDI (Sabatini et al. 2020 – Diversity and Distribution). Even if we cannot be sure a given pixel of deciduous forest actually belongs to the respective PNV class, limiting the comparison to forests within the same PNV class within each region\type combination ensures that the forests being compared have similar environmental conditions (if not composition), therefore strongly reducing the bias highlighted above. While I see the practical advantage of using Corine Land Cover classes, a simple broadleaf vs conifer distinction is way too coarse to account for the ecological differences between (extremely) different growing conditions.

My second point is to a certain extent related to the first. It relates to the way plantations and forests composed by non-native species are considered. At the time being, the ms simply distinguishes between conifer and deciduous forests, not questioning whether the composition of the forest is coherent with the biogeographical context and PNV. For instance, conifer plantations of non-native species are widespread in central Europe. By simply comparing one of these stands to a reference conifer forest in the same biogeographical region there is the risk of returning an overoptimistic picture of the state of these plantations. They might be in ‘good’ conditions when it comes to soil organic carbon, tree cover and so on. But the appropriate reference for that stand is not a native conifer forest (which is likely located in very different ecological conditions), but rather the broadleaf forest that should be there, if we did not replace it with conifers. This makes a big difference in terms of productivity, biodiversity, SOC and so on. Whether it is relatively easy to double check whether a coniferous forest occurs where the PNV is actually broadleaf forest, plantations of broadleaf forests (e.g., Eucalyptus forests) growing where the PNV is, for instance, an oak forest (e.g., Northern Spain) might be harder to discriminate. Yet, I have the impression missing this crucial point might give the false impression that forests in certain regions (e.g., central Europe) are in a much better condition than they actually are. The super positive results obtained for Austria (one of the countries with the highest Fellings as percent of net annual increment – 87.1% based on Indicator 3.1 of Forest Europe 2020).

These are two very pertinent comments which made us reflect on our approach and on the validity of our results. As explained above, we did not want to give up compliance with the accounting framework of using reference data sets that will be regularly updated in the future and kept our forest classification based on Corine Land Cover, which is a standard for Europe. We agree that the distribution of forest over only four classes is quite coarse. The alternative exists, e.g. in Italy where a fourth level of Corine breaks down forests over different classes with dominant tree species, but regularly updated maps of forest tree species or forest habitats that go beyond the four broad classes used in our paper are currently not available. Therefore, we attempted to address the concerns raised here by doing an ex-post uncertainty analysis. The analysis is described in the methods section from L660 onwards and reported in the result section (L113-156). It assigns a level of uncertainty to the forest condition assessment per forest type. The basic idea is that we assign a higher level of uncertainty of the forest condition index for forest types with reference sites with deviating environmental conditions while considering also the potential natural vegetation using the map from Bohn et al.

Some additional points. As a reader, I found the methods difficult to follow, due to the fragmentation between the main text and SI. Honestly, I would give a more thorough description of the data and methods in the main text, and would only refer the reader to the SI for those predictors for which substantial modelling\pre-processing was necessary (e.g., the SOC or biodiversity modelling). The methods part would results substantially longer, but this would also reduce redundancy (some paragraphs are repeated both in the main text and SI).

The editor indicated indeed to us that the methods section has an unlimited word count so we rewrote the methods section and reserved the supplement only for supplementary tables and figures

I don't quite understand the need of section 1 in the SI. Why making the philosophical distinction between forests and forest ecosystems if, due to data constraints (CORINE), forest was treated simply as a land cover class? Is this section really needed?

Fair point. It is no longer part of the revised version

Soil Organic Matter – Please provide more details on how the kriging technique was performed, and the goodness-of-fit evaluated. You state: 'The choice for this method was inspired by extensive testing of various kriging methods on an independent dataset to obtain the best possible outcome. Our results are in line with other interpolations carried out to calculate the value of the SOC in a spatially explicit way13.' I find this statement unconvincing and not transparent. Make sure you give info on the total number of LUCAS soil plots for 2015. Why mentioning the 2009 LUCAS data if not used?

We expanded the description of our method used to interpolate the SOC data on L432-439 and we added supplementary table 4 where we provide correlations between the observed and modelled data

Threatened birds – I'm puzzled by the lack of a spatial component in the model. Ploton et al. 2020 (<https://www.nature.com/articles/s41467-020-18321-y>) explains very well the risk of neglecting the spatial component in these kind of models. Including a spatial component to the GLM would probably change the relative importance of different predictors. I also recommend using spatial cross-validation when checking for model performance.

We don't fully understand this comment about the lack of a spatial component. The birds model is spatial by its nature and fits spatially resolved data of birds. In addition, we included lat/lon coordinates as predictor variables in a first run but they were not retained in the final model. Analysis of spatial autocorrelation in macroecological models is something largely discussed in the literature with no absolute conclusions. For instance, the method of spatial cross validation (Ploton et al. 2020) is criticized in Wadoux et al 2021 (<https://doi.org/10.1016/j.ecolmodel.2021.109692>). In this paper, Wadoux et al clearly state that spatial CV strategies remove entire portions of the geographic and hence also the covariate space, causing under-representation of environmental conditions similar to those at validation locations. This is particularly a problem for biological, ecological and environmental variables which are geographically structured". In this paper repeated k-fold cross validation is also presented as a suitable measure of model accuracy. Complementary, we have performed a repeated 10-fold cross validation using the training dataset (with 100 repetitions) obtaining similar results (mean RMSE=2.13 and mean R2=0.55)." See L498-499, L510-512

Weights for the metrics – Attributing weights is a crucial step of any multi-parametric index. McElhinny et al. 2006 provides some guidance, but there is no real way of ensuring the weights aren't, to a certain

extent, arbitrary. I feel the manuscript should be much more transparent on this, and explain how crucial this step is in the main text, together with a discussion of the strengths and limits of the approach chosen.

We are transparent about the weights. We used supplementary table 9 to construct weights. This table contains 5 conceptual indicator selection criteria that come from the SEEA EA accounting guidelines. Next, we ranked the ecosystem condition indicators giving the highest rank to the indicator that is most relevant for each of the criteria. The sum of the ranks was next converted to a proportion to define the weight. The procedure is explained on L646-650. We performed a sensitivity analysis including the weights to understand their impact on the index. This analysis is reported on L113-126

Line-to-line comments:

TITLE – From the title alone it's unclear what 'standard' the authors are referring to. What about replacing 'global statistical standard' with 'global statistical accounting standard'?

We can replace with international statistical standard

L42 – "Above 25%" change to "more than 25%"

No longer in the revision

L42-44 – I'd include other examples of degradation beside defoliation (e.g., soil erosion, biodiversity loss, decreased resilience to disturbance).

Also this part was not retained in the revised version

L63 – "The conditions to develop such a method are ideal". Unclear. Please rephrase.

No longer part of the text

L95 – It should be 'Increase by 95%', I think (I'm not a native speaker)

changed

L100 – I think a simple t-test is inadequate given the intrinsically spatial (=autocorrelated) nature of the data. A GLMM with a spatial component is probably more adequate here.

We opted for a Man-Whitney U test on a random selection of gridcells in this version (after Jacob et al.; 2014, <https://doi.org/10.1007/s10113-013-0499-2>). L85-91 report the results; Supplementary table 2 reports the test results.

L100 "than in other the forest classes" . Delete 'the' –

deleted

L117 – "A SEEA EA compliant forest condition account...". Convolved sentence. Please revise.

The sentence is split in two

L148-149 – “(average forest condition 2018 = 0.53)”. Unclear whether this refers to ‘Mediterranean forests’. If yes, why not reporting also the average for the other forest types listed? How do these values relate to those reported at L154-155.

Thank you for reporting this mistake. The caption is now streamlined

L204 – ‘well connected’ – Honestly, I don’t find a FAD metric a very convincing indicator of connectivity. In the same 23x23 moving window, the same amount of forest pixels might be interspersed and disconnected, or all clustered in a connected, large patch, without the FAD distinguishing the two situations. I would avoid overselling these results, and maybe delete the whole paragraph L204-208.

We appreciate your point and deleted that section from the paper. We note that FAD is still used by reporting authorities such as FAO, ForestEurope and Eurostat; an alternative that addresses your point is under submission to a journal.

L219 – “...productivity in the boreal.” Add ‘forest’ after ‘boreal’.

We used region instead of forest to avoid having forest twice.

L253 – The MAIN limitation, i.e., the low representativity of the reference dataset for the forests in each region\forest type combination is not even mentioned here.

We address this now in more detail in L265-296

Reviewer #2 (Remarks to the Author):

The study by Maes et al. examines the condition of forests in Europe and investigates the changes from 2000 to 2018. For this purpose, an index was determined, which is based on seven ecosystem variables like tree cover density. The authors have prepared these variables for the whole of Europe, mainly from established satellite products, with great effort.

However, it is not entirely clear to me what is really novel and innovative about this study. It seems that the basic method has already been published. So, this study is a pure application of a known method to all forests of Europe. It must be mentioned that similar products have already been published by other scientists (which, to be fair, the authors themselves also state L77-78, L80-82). I would recommend to highlight more what is special about your study, besides mentioning 'operationalizes a method' (L83) and 'first, large scale test' (L89). The methods are clearly described and the figures are mostly comprehensible. The statistics for the calculation are rather simple, which I very much welcome, because it makes the calculation easy to understand.

We understand your point. Then again, the study is unique in testing the SEEA-EA framework at continental level, and comparing the condition of forests against a natural reference (as far as we know this comparison has never been done before). We see added value of using the ecosystem accounting system, as explained also to the first reviewer.

Detailed comments:

- The title and text often refer to "global statistical standard". What is that supposed to be? This is not a common term in statistics. For the title it is maybe not needed to mention.

We can propose using "international statistical standard". The UN statistical commission adopted on 21 March 2021 the SEEA EA (and in particular the chapter on ecosystem condition accounting) as "international statistical standard" (see: <https://unstats.un.org/unsd/statcom/52nd-session/documents/decisions/Draft-Decisions-Final-5March2021.pdf>; Decision 8, c).

- The introduction is well written (perhaps a bit too long), however the references are mainly reports or book chapters (ref 1-3,6,8,10-18). These are sometimes not clear to find (no links were given) and it is not clear to me if these are peer-reviewed. I find this unusual. I would like to see more state-of-the-art literature, which is definitely listed later in the Discussion.

We rewrote the introduction with more attention to the peer reviewed literature.

- It is not clear in the main text what the Europe-wide forest condition index is based on. It is described in detail in the appendix. However, it would be helpful to have a sense of what data goes into the index before reading the results. This should be mentioned briefly (L84 'spatial data').

All the methods are now described in the method section but the journal format places that section after the results. To accommodate this comment, we included a few lines in the introduction L51-56 to increase readability.

- In addition, the selection of the reference levels (min and max) has a strong influence on the final index. It would be advisable to see some sort of sensitivity analysis on the assumptions made. However, it should not affect the analysis of the changes.

We now included a parameter sensitivity analysis

- L97-L101: I find the temporal changes in the index (from 0.57 in 2000 to 0.58 in 2018) to be very small. I was surprised that this change should be significant, considering the high standard deviations of the indicator values given in Table 1 (last two columns – SD between 0.12 and 0.19).

In the first manuscript, we did a t-test for dependent samples to assess if forest condition changes significantly across forest type, not within forest type. So we compared the average forest condition in 2000 and 2018 using the already averaged condition of 44 forest types. This is clearly confusing and we abandoned the test. Instead, we present in Figure 2 the changes in forest condition per forest type. Statistical testing is done only per forest type comparing for each type separately the condition in 2000 and 2018. The results are reported in Supplementary table 2. We did not add error bars to figure 2 to increase the readability.

- L103-104: 'indicators ... between 0.6 and 0.8'. Where do I see that? Table 1 shows indicator values between 0.2 and 0.8. Right?

We moved table 1 to the supplement and replaced it with Figure 3 to show the value of indicators relative to the scale of measurement and relative to the lower and upper reference levels. Hopefully this is clearer.

- L105: species richness reached 0.48 in 2018 (not 0.52), right?

Thank you for pointing this out. There were some errors in the rescaled indicator values in the table. This is now corrected.

- L108-113: Where do I see the results to these statements, in Figure 1?

We have now included conditions for 2000 and 2018 per forest type in Figure 2. The lines you refer to are now L105-112. We indeed do not provide numbers as the text describes the general patterns that should emerge from Figure 2 if you would read it horizontally by region or vertically by forest class. The average condition values could be added. We did not want to overload the paper with averages based on averages.

- Table 1: How have you calculated the indicator values? For example, SOC: the reference levels are 0 – 0.63. If transforming the 2000-value 0.09 it should be 14.3% and not 20%. Or do I have a mistake in thinking?

This is correct. See also the above comment. The variable values and reference levels are correct but you spotted indeed an error in the calculation. Thank you for being so sharp on this. It is slightly embarrassing that this passed in the previous version without any of us noting.

- Table 1: I would recommend to add the final forest condition index in this table including also the sd.

Also this table disappeared but we recall that the paper comes with spreadsheet and a raster that allows readers to aggregate the data in different ways. We also included in Supplementary figure 1 the data distribution of the forest condition index 2000 and 2018 by forest type.

- Figure 3: I'm really sorry, but I don't fully understand this figure. What do the dots stand for? Are they individual countries or regions?

We removed this entire analysis from the paper. We will likely submit this elsewhere as it needs more detailed descriptions.

- L209: highest indicator value for 2018 in NDVI. I am surprised, because 2018 was particularly dry and therefore the forests probably less productive. What has this high indicator value then for an interpretation?

In 2018 annual precipitation was close to average; however the north of Europe experienced a very dry year and the south a very wet year. Soil moisture was indeed quite low. See the Copernicus climate bulletin for 2018: <https://climate.copernicus.eu/european-wet-and-dry-conditions>

To smooth annual variability, our NDWI and NDVI based indicators use a three-year moving average however.

Reviewer #3 (Remarks to the Author):

General comments

The article presents a comprehensive assessment of ecosystem condition of European forests based on SEEA EA methodology. Without any doubt, the study provides valuable contribution to the application and development of SEEA EA. European states will be increasingly searching inspiration and methodological guidance for condition assessment in ecosystem accounting.

As authors admit, there is available relatively extensive scholarship on the assessment of forest condition in Europe. The major innovation is computation of forest condition index based on seven condition variables. However, the procedure of selection and calculation of the index would require further attention.

Especially regarding the description of assigning the weights and uncertainty analysis of the index. While authors declare that the change of the condition from 0.57 to 0.58 in the period 2000 – 2018 is statistically significant, the uncertainty range associated with this change is not reported.

Here we opted for a different statistical approach (see also our earlier responses) but the assessment by forest type still reveals statistically significant differences. In short, we now used a Mann-Whitney-U test on a random selection of grid cells to test whether or not average condition by forest type is significantly different between 2000 and 2018.

The general comment related to the System of Environmental-Economic Accounting – Ecosystem Accounting (SEEA-EA). Authors are focusing primarily on ecosystem condition but present also data on ecosystem extent. As the SEEA EA system consists of several mutually related types of accounts, including extent, condition, ecosystem services supply (biophysical and monetary) and monetary asset account, authors should better position their work in this framework. Rather than technical discussion of particular parameters, it would be beneficial to discuss more context-relevant possibilities for application of condition assessments at the European level and national states.

We agree. We attempted to better frame SEEA EA in the introduction (L34-50); reduced in the revised version the discussion on the individual indicators and highlighted in the discussion under the section Strengths and applications (L311 and further) the relevance of ecosystem condition accounts for EU policy making

Authors selected a top-down indicators approach based on European-wide datasets. Therefore, the study builds on available data to support SEEA Ecosystem Accounting. Moreover, condition assessment is based on the distance to the reference condition, i.e. primary intact or protected forest. What is the theoretical framework for the assumption of the good condition only in a natural state? What is the role of the forest management for the condition status when delivering various ecosystem services? These important questions has not been addressed by authors.

The theoretical foundation of the study is based in the notion of ecosystem condition adopted in the SEEA-EA framework. Which is developed in Keith et al (2020, <https://doi.org/10.3897/oneeco.5.e58216>) and in Winter et al. (2010; 2012, <https://doi.org/10.1093/forestry/cps004>; <https://doi.org/10.1016/j.foreco.2010.01.040>). Ecosystem condition in the SEEA-EA is the quality of an ecosystem measured in terms of its abiotic and biotic characteristics. Within this framework, integrity is

used as a comparison benchmark for measuring condition (reference sites). However, integrity is not considered a surrogate of condition. In general, forest management oriented to wood production produces a gradient of forest (stands) with different levels of naturalness at landscape level. That is, from low intensity managed forest, e.g. close-to-nature forestry, to high intensity managed forest, e.g. anthropogenic exotic plantations. Reasonably, the distance to reference sites is expected to differ depending on the level of modification introduced by forest managers, among other drivers. In consequence, managed forest can exhibit different levels of condition across the gradient. Therefore, not all managed forest are necessarily in bad condition across a gradient of management intensity. There is a range of forest management approaches oriented to wood production that can still deliver forest in good condition and a series of ecosystem services beyond timber.

In the paper we now recommend in lines 298 – 301 that SEEA EA guidance could be more specific on how to factor in forest management. This is a lesson learnt based on the European context. Indeed, the current guidelines suggest to assess the condition of ecosystems versus a natural or anthropogenic baseline. In reality, there is a gradient from natural to plantation which is insufficiently captured by the accounting rules. So although we address this briefly in the revised version, space limitations limit us from a more in-depth discussion. We decided to dedicate a separate manuscript on this issue.

Related is the logic behind the indicators selection and weighting. The selection of indicators and assigning of the weights is not sufficiently substantiated by authors. For the analysis to be sufficiently transparent, this should be explicitly addressed and described.

Also noted by the first reviewer. We used supplementary table 9 to construct weights. This table contains 5 conceptual indicator selection criteria that come from the SEEA EA accounting guidelines. Next we ranked the ecosystem condition indicators giving the highest rank to the indicator that is most relevant for each of the criteria. The sum of the ranks was next converted to a proportion to define the weight. The procedure is explained on L703-711. We performed a sensitivity analysis including the weights to understand their impact on the index. This analysis is reported on L113-126

Finally, authors apply SEEA EA approach. What are the lessons learnt and recommendation for future applications of ecosystem condition accounting? Authors list strengths connected to nature restoration targets and global analysis, but it could be interesting to see recommendations or evaluation for the SEEA EA accounting itself. Are there some obstacles or what the future directions are?

We included a short section now with 3 main recommendations for improved guidance on forest accounts (L298-310). We also rewrote the section on strengths and applications (L311 and further).

Specific comments for the article (line no. indicated):

Title: Authors are analysing a condition and its change in European forests. It would be more accurate to incorporate change into the title. Maybe a focus just on recent period (2018) would lead to more straightforward analysis than comparing only 2 terms.

We now changed the title (admittedly with some mild reluctance of the first author) to include the word “Accounting” as a way to incorporate change. The strength of accounting lies really in the provision of regularly updated data so hence the revised version also reports 2000 and 2018 results.

91-92: Authors are focusing on ecosystem condition but here they report changes in ecosystem extent. Ecosystem extent changes are presented in Supplement. It would be good to align extent results with the whole article, either incorporate in main text or move completely to Supplement. (Also in the context of extensive discussion of extent change in lines 192-203).

We removed most of the results on extent but we kept the extent account as a supplement. This is relevant for interpretation of the results, for scientists who want to make use of our datasets, or for assessments of ecosystem services as a function of extent and condition.

97-100: Average ecosystem condition is presented without uncertainty range, so the reader is not guided what is the possible range of the index.

True. In defense, we provide the 100 m condition grids and we also include now a supplementary figure 1 with condition histograms by forest type. Also the comparison between forest condition 2000 and 2018 by forest type comes with a table of statistical significance testing; this is reported in supplementary table 2.

150: Spatial mean condition is 0.622. While average condition for forest types is 0.58. It is clear that this is the outcome of computation approach, but authors could maybe report this in main results and/or align these two numbers.

We removed reference to the 0.622, mainly to avoid confusion. We prefer reporting as much as possible on the level of forest types and we avoid comparing averages across forest types with spatial averages. We recall that the datasets including the 100 m condition grids will be made publicly available so that users can also aggregate the data by other spatial typologies.

158: Comparing condition with Habitats condition: this part looks disconnected to the overall assessment. It could be one of the main outputs, but this would require restructuring the article and elaborating more on the prerequisites of the assessment.

We removed this comparison from the paper in favour of an uncertainty assessment. We intend to publish the comparison between forest condition and forest habitat conservation status as a separate paper.

161-164: What was the aim of determining a threshold value? It is not fully understandable, how the threshold values was derived.

Also this comment is not addressed as we removed the analysis from the paper

204 – 205: Repetition of statement from lines 185-187.

This section is removed from the paper (also in response to reviewer 1)

254: Selection of forest condition indicators. Authors give a rationale for the overall selection of indicators measuring ecosystem condition. They selected 7 indicators. However, some of the commonly used indicators are intentionally excluded. For example, defoliation or common forest bird species. Selection of indicators influences final index values and the sensitivity to indicator selection is not known. That's one of the problematic points of the study. What is the effect of other indicators? If other metrics were included, would be overall condition index different? Authors illustrate this by critical loads

exceedance but this discussion is not generally applicable (of course, it would be preferable to use state rather than pressure indicators).

We added one more sentence on the indicator selection on L398: (“This selection stems from a broader list of 59 forest condition indicators that have been agreed with EU member states and experts to map and assess forest condition under Action 5 of the EU Biodiversity Strategy to 2020”). I appreciate much the comment but to save space we limit this part.

In short: this paper is not a stand alone exercise. In 2010 the European Commission, as part of its biodiversity strategy, set up an initiative to create a knowledge base on ecosystems. This initiative on mapping and assessment of ecosystems and their services (MAES) was a collaborative effort with EU countries and scientific experts to develop typologies and indicators for ecosystem, ecosystem condition and ecosystem services. On condition, we consulted in 2017 over 90 experts in several workshops to establish a list of common indicators to analyse pressures on ecosystems and ecosystem condition (<https://data.europa.eu/doi/10.2779/055584>). This list was the basis for an EU wide ecosystem assessment including forests (<https://data.europa.eu/doi/10.2760/519233>). Finally we applied the SEEA EA condition indicator guidelines to this list to end up with the final selection of seven forest condition indicators. Furthermore, the sensitivity analysis tackles some of your points and illustrates the robustness of the forest condition index.

300: Authors state that presented method can help member states report on Habitats Directive. However, as this method was developed to support SEEA EA, I would appreciate elaboration of this proposal with regard to current reporting obligations. Would really member states report ecosystem condition in the same way within SEEA EA and Habitats Directive? See also the point above.

As the section was removed, we suggest not to dwell on this particular issue

333: Authors use intersection of Corine land cover and biogeographic regions for the classification of forest types. This approach provides a meaningful overview at the European level. However, it would be beneficial to discuss implications of this approach for the assessment of forest condition with regard to other classifications.

We used the map of potential natural vegetation for assessing the level of correspondence with the forest types. This resulted in a reasonable level of agreement between both classifications (Supplementary Figure 3). The comparison of the classification of forest types vs the map of potential natural vegetation is now documented in the supplementary material. In addition, the map of potential natural vegetation was integrated in an uncertainty assessment identifying for each forest type the degree of uncertainty on the basis of four subjects described in Table 2.

395: Partial repetition of text from lines 160 – 170.

This is also removed

Supplementary information

58 – 66: Forest typology was derived from the intersection of Corine Land Cover and biogeographic regions. The resulting classification reflects broader regions. However, it is not able to discern specific habitat types. For example, continental broad-leaved forests still incorporate large diversity of forest

stands, regarding species composition, forest types and management. How this relatively coarse approach could influence resulting condition values?

This comment partially overlaps with the comments of the first reviewer. Whereas we explained earlier that we prefer keeping the current typology based on 44 classes, we hope that the new uncertainty analysis does address these issues. There is still a shortage of regularly updated data on tree species occurrence or forest management, let alone that baseline data are available at EU level. We are convinced that the combination of Corine Land Cover with the biogeographical regions is currently the best available data to assess changes in forest extent and condition over time. We agree that the last few years the shift to high resolution sensors improved forest statistics but many products are still one-off and do not allow accounting.

94: The methodology behind the selection of indicators is not explicitly described. What was the exact selection procedure? Literature search, expert opinion, etc.? This should be transparently described.

This is a valid point. But we prefer to simply refer on line 381-383 to our earlier work in this respect. The development of a forest condition account is based on earlier work and research under Action 5 of the EU strategy to 2020. Under that action, EU countries were required to Map and Assess Ecosystems and their Services on their territory. This so-called MAES initiative collected the experiences within EU countries and the knowledge of experts to develop an ecosystem typology and indicator frameworks for ecosystem services and ecosystem condition. A specific guidance document for condition defined a list of indicators per ecosystem type (<https://op.europa.eu/en/publication-detail/-/publication/42d646b6-1c3a-11e8-ac73-01aa75ed71a1/language-en/format-PDF/source-282683921>). For forests a list with 59 condition indicators (and 21 pressure indicators) was agreed with member states and experts for reporting forest condition. This list was also the basis for an EU ecosystem assessment delivered in 2020 (<https://op.europa.eu/en/publication-detail/-/publication/afac1162-0f58-11eb-bc07-01aa75ed71a1/language-en>) and was finalized in the recently published EU methodology to map and assess ecosystem condition (<https://data.europa.eu/doi/10.2760/13048>). Our paper builds on this earlier work but aligned the MAES condition typology with the one from SEEA EA and reduced the number of indicators mainly due to data availability. The requirement of spatially explicit, regularly updated datasets for accounting constrained the number of indicators.

429- Table S7: The same remark can apply to weight selection. How were weights determined, respectively how was the importance of the indicators assessed?

We repeat here our earlier reply: We used supplementary table 9 to construct weights. This table contains 5 conceptual indicator selection criteria that come from the SEEA EA accounting guidelines. Next we ranked the ecosystem condition indicators giving the highest rank to the indicator that is most relevant for each of the criteria. The sum of the ranks was next converted to a proportion to define the weight. The procedure is explained on L634-638. We performed a sensitivity analysis including the weights to understand their impact on the index. This analysis is reported on L116-129

451: When comparing condition index to EU Habitats status, authors should discuss the data entering this assessment. To which extent it's harmonized among member states and what types of measurements are entering this assessment?

As the section was removed, we suggest not to dwell on this particular issue

REVIEWERS' COMMENTS

Reviewer #1 (Remarks to the Author):

The authors did a good job at addressing my previous concerns, and I very much appreciate both the sensitivity and the uncertainty analyses they added to the manuscript. I think the ms is now better balanced at conveying its main results, while acknowledging the intrinsic limitations of making such an assessment at continental level. Honestly, I was hoping to see these limitations better integrated in the accounting (e.g., giving less weight to uncertain forest types when summarizing results at the continental or biogeographical region level). I do realize, however, that there is no straightforward way to do it.

I have a few remaining minor comments, which the authors can find in the attached, commented, pdf. The main three are clarifications requests: 1) Could you please clarify how the bird data were used to estimate species richness at year 2000, and year 2018? 2) Could you please provide some more detail on the SEEA-EA conceptual criteria used to weight the seven indicators? 3) How were the four uncertainty criteria shown in Table 2 summarized at forest type level?

Reviewer #2 (Remarks to the Author):

The authors have invested a lot in introducing new aspects, adding new figures, results and explanations; some parts have also been removed in order to better guide the reader through the manuscript. The new figures are much easier to understand (Fig.2+3), and the uncertainty analysis for primary/protected forests helps to evaluate the results. The introduction has been substantially revised and more peer-reviewed state-of-the-art literature has been added (instead of reports). Basically, all my comments have been satisfactorily answered and implemented.

More general comment on the revision: where do I see the changes in the manuscript compared to the first version? What do the yellow marked text passages mean? The changes made to the text are not traceable, because even in places not marked in yellow, sometimes extensive changes have been made. In this case, I recommend a document with track changes to make it easier for reviewers to identify the revisions.

A response to a longer question about the methodology: "We have now included a parameter sensitivity analysis" - where can I find this? What are the results of this analysis and what is the answer to my comment? I have searched it in the text and in the SI, but it is very time-consuming to find the relevant sections and figures. Anyway, from my point of view they have done a reasonable sensitivity analysis (e.g. SI Figure 2), which often shows small changes in the Forest Condition Index, and it is encouraging that according to this study the upper and lower limits have almost no influence. However, the index responds to 10% changes in the SOC weight (-8.14%) and in the Landscape Naturalness weight (+5.66%). The authors consider this result to be robust. However, if you change the weight of a single quantity by 10%, I find that a change of more than 1-2% in the final index is already not robust. I would like to see a bit more discussion of these sensitivity results, as the weights of these variables seem to have an impact on the results, but the choice of weight values is not straightforward.

Reviewer #3 (Remarks to the Author):

Thanks to authors for detailed responses on the first review.

I have just few additional remarks.

Assessing the change: authors state that their framework enables reporting and analysis of past trends (line 60). They compare years 2000 and 2018. In first review, I expressed a concern that comparing just two time periods is not sufficient to analyse change. What is the reason authors did not included other time horizons (2006, 2012). Was it influenced by data availability on condition or other factors?

Results: I would suggest starting with more general patterns and "bigger" trends in Europe and continue with specific regions etc. Currently, the style of presenting results is not consistent, going to and fro about main trends, regional specifics, trends in improvement etc. This could be really done better.

I would carefully consider use of technical details in results which should be rather addressed in methods section.

E.g. line 95: Figure 3 also provides values that are rescaled between 0 and 1 and which are used to calculate the forest condition index.

Line 119: We evaluated the sensitivity of the forest condition index to changes of these parameters by recalculating the index while perturbing the parameters one by one with 10% on their scale of measurement.

And some others. These are redundant in results, according to my view, and could be addressed elsewhere.

Otherwise, article improved and thanks to authors for their effort.